# LAST-ITERATE CONVERGENCE RATES FOR MIN-MAX OPTIMIZATION

## ABSTRACT

While classic work in convex-concave min-max optimization relies on average-iterate convergence results, the emergence of nonconvex applications such as training Generative Adversarial Networks has led to renewed interest in last-iterate convergence guarantees. Proving last-iterate convergence is challenging because many natural algorithms, such as Simultaneous Gradient Descent/Ascent, provably diverge or cycle even in simple convex-concave min-max settings, and previous work on global last-iterate convergence rates has been limited to the bilinear and convex-strongly concave settings. In this work, we show that the HAMILTONIAN GRADIENT DESCENT (HGD) algorithm achieves linear convergence in a variety of more general settings, including convex-concave problems that satisfy a novel "sufficiently bilinear" condition. We also prove convergence rates for stochastic HGD and for some parameter settings of the Consensus Optimization algorithm of Mescheder et al. (2017).

## 1 INTRODUCTION

In this paper we consider methods to solve smooth unconstrained min-max optimization problems. In the most classical setting, a min-max objective has the form

$$\min_{x_1} \max_{x_2} g(x_1, x_2)$$

where $g : \mathbb{R}^d \times \mathbb{R}^d \to \mathbb{R}$ is a smooth objective function with two inputs. The usual goal in such problems is to find a saddle point, also known as a *min-max solution*, which is a pair $(x_1^*, x_2^*) \in \mathbb{R}^d \times \mathbb{R}^d$ that satisfies

$$g(x_1^*, x_2) \leq g(x_1^*, x_2^*) \leq g(x_1, x_2^*) \tag{1}$$

for every $x_1 \in \mathbb{R}^d$ and $x_2 \in \mathbb{R}^d$. Min-max problems have a long history, going back at least as far as Neumann (1928), which formed the basis of much of modern game theory, and including a great deal of work in the 1950s when algorithms such as *fictitious play* were explored (Brown, 1951; Robinson, 1951).

The *convex-concave* setting, where we assume $g$ is convex in $x_1$ and concave in $x_2$, is a classic min-max problem that has a number of different applications, such as solving constrained convex optimization problems. While a variety of tools have been developed for this setting, a very popular approach within the machine learning community has been the use of so-called *no-regret algorithms* (Cesa-Bianchi & Lugosi, 2006; Hazan, 2016). This trick, which was originally developed by Hannan (1957) and later emerged in the development of boosting (Freund & Schapire, 1999), provides a simple computational method via repeated play: each of the inputs $x_1$ and $x_2$ are updated iteratively according to no-regret learning protocols, and one can prove that the average-iterates $(\bar{x}_1, \bar{x}_2)$ converge to a min-max solution.

Recently, interest in min-max optimization has surged due to the enormous popularity of Generative Adversarial Networks (GANs), whose training involves solving a nonconvex min-max problem where $x_1$ and $x_2$ correspond to the parameters of two different neural nets (Goodfellow et al., 2014). The fundamentally nonconvex nature of this problem changes two things. First, it is infeasible to find a "global" solution of the min-max objective. Instead, a typical goal in GAN training is to find a local min-max, namely a pair $(x_1^*, x_2^*)$ that satisfies (1) for all $(x_1, x_2)$ in some neighborhood of $(x_1^*, x_2^*)$.

Second, iterate averaging lacks the theoretical guarantees present in the convex-concave setting. This has motivated research on *last-iterate* convergence guarantees, which are appealing because they more easily carry over from convex to nonconvex settings.

Last-iterate convergence guarantees for min-max problems have been challenging to prove since standard analysis of no-regret algorithms says essentially nothing about last-iterate convergence. Widely used no-regret algorithms, such as Simultaneous Gradient Descent/Ascent (SGDA), fail to converge even in the simple *bilinear* setting where $g(x_1, x_2) = x_1^\top C x_2$ for some arbitrary matrix $C$. SGDA provably cycles in continuous time and diverges in discrete time (see for example Daskalakis et al. (2018); Mescheder et al. (2018)). In fact, the full range of Follow-The-Regularized-Leader (FTRL) algorithms provably do not converge in zero-sum games with interior equilibria (Mertikopoulos et al., 2018). This occurs because the iterates of the FTRL algorithms exhibit cyclic behavior, a phenomenon commonly observed when training GANs in practice as well.

Much of the recent research on last-iterate convergence in min-max problems has focused on *asymptotic* or *local* convergence (Mertikopoulos et al., 2019; Mescheder et al., 2017; Daskalakis & Panageas, 2018; Balduzzi et al., 2018; Letcher et al., 2019; Mazumdar et al., 2019). While these results are certainly useful, one would ideally like to prove *global non-asymptotic* last-iterate convergence rates. Provable global convergence rates allow for quantitative comparison of different algorithms and can aid in choosing learning rates and architectures to ensure fast convergence in practice. Yet despite the extensive amount of literature on convergence rates for convex optimization, very few global last-iterate convergence rates have been proved for min-max problems. Existing work on global last-iterate convergence rates has been limited to the bilinear or convex-strongly concave settings (Tseng, 1995; Liang & Stokes, 2019; Du & Hu, 2019; Mokhtari et al., 2019). In particular, the following basic question is still open:

"What global last-iterate convergence rates are achievable for convex-concave min-max problems?"

**Our Contribution** Understanding global last-iterate rates in the convex-concave setting is an important stepping stone towards provable last-iterate rates in the nonconvex-nonconcave setting. Motivated by this, we prove new linear last-iterate convergence rates in the convex-concave setting for an algorithm called HAMILTONIAN GRADIENT DESCENT (HGD) under weaker assumptions compared to previous results. HGD is gradient descent on the squared norm of the gradient, and it has been mentioned in Mescheder et al. (2017); Balduzzi et al. (2018). Our results are the first to show non-asymptotic convergence of an efficient algorithm in settings that not linear or strongly convex in either input. In particular, we introduce a novel "sufficiently bilinear" condition on the second-order derivatives of the objective $g$ and show that this condition is sufficient for HGD to achieve linear convergence in convex-concave settings. The "sufficiently bilinear" condition appears to be a new sufficient condition for linear convergence rates that is distinct from previously known conditions such as the Polyak-Łojasiewicz (PL) condition or pure bilinearity. Our analysis relies on showing that the squared norm of the gradient satisfies the PL condition in various settings. As a corollary of this result, we can leverage Karimi et al. (2016) to show that a stochastic version of HGD will have a last-iterate convergence rate of $O(1/\sqrt{k})$ in the "sufficiently bilinear" setting. On the practical side, while vanilla HGD has issues training GANs in practice, Mescheder et al. (2017) show that a related algorithm known as Consensus Optimization (CO) can effectively train GANs in a variety of settings, including on CIFAR-10 and celebA. We show that CO can be viewed as a perturbation of HGD, which implies that for some parameter settings, CO converges at the same rate as HGD.

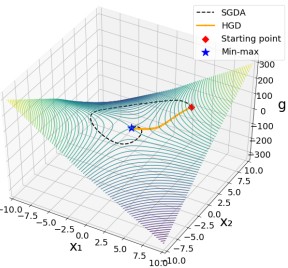

Figure 1: HGD converges quickly, while SGDA spirals. This nonconvex-nonconcave objective is in defined in Appendix K.

We begin in Section 2 with background material and notation, including some of our key assumptions. In Section 3, we discuss Hamiltonian Gradient Descent (HGD), and we present our linear convergence rates for HGD in various settings. In Section 4, we present some of the key technical components used to prove our results from Section 3. Finally, in Section 5, we present our results for Stochastic HGD and Consensus Optimization. The details of our proofs are in Appendix H.

## 2 BACKGROUND

### 2.1 PRELIMINARIES

In this section, we discuss some key definitions and notation. We will use $||\cdot||$ to denote the Euclidean norm for vectors or the operator norm for matrices or tensors. For a symmetric matrix $A$, we will use $\lambda_{\min}(A)$ and $\lambda_{\max}(A)$ to denote the smallest and largest eigenvalues of $A$. For a general real matrix $A$, $\sigma_{\min}(A)$ and $\sigma_{\max}(A)$ denote the smallest and largest singular values of $A$.

**Definition 2.1.** *A critical point of $f : \mathbb{R}^d \to \mathbb{R}$ is a point $x \in \mathbb{R}^d$ such that $\nabla f(x) = 0$.*

**Definition 2.2** (Convexity / Strong convexity). *Let $\mu \geq 0$. A function $f : \mathbb{R}^d \to \mathbb{R}$ is $\mu$-strongly convex if for any $u, v \in \mathbb{R}^d, f(u) \geq f(v) + \langle \nabla f(v), u - v \rangle + \frac{\mu}{2} ||u - v||$. When $f$ is twice-differentiable, $f$ is $\mu$-strongly-convex iff for all $x \in \mathbb{R}^d, \nabla^2 f(x) \succeq \mu I$. If $\mu = 0$ in either of the above definitions, $f$ is called convex.*

**Definition 2.3** (Monotone / Strongly monotone). *Let $\mu \geq 0$. A vector field $v : \mathbb{R}^d \to \mathbb{R}^d$ is $\mu$-strongly monotone if for any $x, y \in \mathbb{R}^d, \langle x - y, v(x) - v(y) \rangle \geq \mu ||x - y||^2$. If $\mu = 0$, $v$ is called monotone.*

**Definition 2.4** (Smoothness). *A function $f : \mathbb{R}^d \to \mathbb{R}$ is $L$-smooth if $f$ is differentiable everywhere and for all $u, v \in \mathbb{R}^d$ satisfies $||\nabla f(u) - \nabla f(v)|| \leq L ||u - v||$.*

**Notation**  Since $g$ is a function of $x_1 \in \mathbb{R}^d$ and $x_2 \in \mathbb{R}^d$, we will often consider $x_1$ and $x_2$ to be components of one vector $x = (x_1, x_2)$. We will use superscripts to denote iterate indices. Following Balduzzi et al. (2018), we use $\xi = (\nabla_{x_1} g, -\nabla_{x_2} g)$ to denote the signed vector of partial derivatives. Under this notation, the Simultaneous Gradient Descent/Ascent (SGDA) update can be written as $x^{(k+1)} = x^{(k)} - \eta \xi(x^{(k)})$.

We will use $J$ to denote the Jacobian of $\xi$, i.e. $J \equiv \nabla \xi = \begin{pmatrix} \nabla^2_{x_1 x_1} g & \nabla^2_{x_1 x_2} g \\ -\nabla^2_{x_2 x_1} g & -\nabla^2_{x_2 x_2} g \end{pmatrix}$. Note that unlike the Hessian in standard optimization, $J$ is not symmetric, due to the negative sign in $\xi$. When clear from the context, we often omit dependence on $x$ when writing $\xi, J, g, \mathcal{H}$, and other functions. Note that $\xi, J$, and $\mathcal{H}$ are defined for a given objective $g$ – we omit this dependence as well for notational clarity. We will always assume $g$ is sufficiently differentiable whenever we take derivatives. In particular, we assume second-order differentiability in Section 3.

We will also use the following non-standard definition for notational convenience:

**Definition 2.5** (Higher-order Lipschitz). *A function $g : \mathbb{R}^d \to \mathbb{R}$ is $(L_1, L_2, L_3)$-Lipschitz if for all $x \in \mathbb{R}^d, ||\xi(x)|| \leq L_1$ and $||\nabla \xi(x)|| \leq L_2$, and for all $x, y \in \mathbb{R}^d, ||\nabla \xi(x) - \nabla \xi(y)|| \leq L_3 ||x - y||$.*

**Notions of convergence in min-max problems**  The convergence rates in this paper will apply to min-max problems where $g$ satisfies the following assumption:

**Assumption 2.6.** *All critical points of the objective $g$ are global min-maxes (i.e. they satisfy (1)).*

In other words, we prove convergence rates to min-maxes in settings where convergence to critical points is necessary and sufficient for convergence to min-maxes. This assumption is true for convex-concave settings, but also holds for some nonconvex-nonconcave settings, as we discuss in Appendix E. This assumption allows us to measure the convergence of our algorithms to $\epsilon$-*approximate critical points*, defined as follows:

**Definition 2.7.** *Let $\epsilon \geq 0$. A point $x \in \mathbb{R}^d \times \mathbb{R}^d$ is an $\epsilon$-approximate critical point if $||\xi(x)|| \leq \epsilon$.*

Convergence to approximate critical points is a necessary condition for convergence to local or global minima, and it is a natural measure of convergence since the value of $g$ at a given point gives no information about how close we are to a min-max. Our main convergence rate results focus on this first-order notion of convergence, which is sufficient given Assumption 2.6. We discuss notions of second-order convergence and ways to adapt our results to the general nonconvex setting in Appendix A.

## 2.2 RELATED WORK

**Asymptotic and local convergence**  Several recent papers have given asymptotic or local convergence results for min-max problems. Mertikopoulos et al. (2019) show that the *extragradient* (EG) algorithm converges asymptotically in a broad class of problems known as coherent saddle point problems, which include quasiconvex-quasiconcave problems. However, they do not prove convergence rates. For more general smooth nonconvex min-max problems, a number of different papers have given local stability or local asymptotic convergence results for various algorithms, which we discuss in Appendix A.

**Non-asymptotic convergence rates**  Work on global non-asymptotic last-iterate convergence rates has been limited to very restrictive settings. A classic result by Rockafellar (1976) shows a linear convergence rate for the proximal point method in the bilinear and strongly convex-strongly concave cases. Another classic result, by Tseng (1995), shows a linear convergence rate for the extragradient algorithm in the bilinear case. Liang & Stokes (2019) show that a number of algorithms achieve a linear convergence rate in the bilinear case, including Optimistic Mirror Descent (OMD) and Consensus Optimization (CO). They also show that SGDA obtains a linear convergence rate in the strongly convex-strongly concave case. Mokhtari et al. (2019) show that OMD and EG obtain a linear rate for the strongly convex-strongly concave case, in addition to proving similar results for generalized versions of both algorithms. Finally, Du & Hu (2019) show that SGDA achieves a linear convergence rate for a convex-strongly concave setting with a full column rank linear interaction term.[1]

**Non-uniform average-iterate convergence**  A number of recent works have studied the convergence of non-uniform averages of iterates, which can be viewed as an interpolation between the standard uniform average-iterate and last-iterate. We discuss these works further in Appendix B.

## 3  HAMILTONIAN GRADIENT DESCENT

Our main algorithm for finding saddle points of $g(x_1, x_2)$ is called HAMILTONIAN GRADIENT DESCENT (HGD). HGD consists of performing gradient descent on a particular objective function $\mathcal{H}$ that we refer to as the *Hamiltonian*, following the terminology of Balduzzi et al. (2018).[2] If we let $\xi := \left( \frac{\partial g}{\partial x_1}, -\frac{\partial g}{\partial x_2} \right)$ be the vector of (appropriately-signed) partial derivatives, then the Hamiltonian is:

$$\mathcal{H}(x) := \tfrac{1}{2}\|\xi(x)\|^2 = \tfrac{1}{2} \left( \|\tfrac{\partial g}{\partial x_1}(x)\|^2 + \|\tfrac{\partial g}{\partial x_2}(x)\|^2 \right).$$

Since a critical point occurs when $\xi(x) = 0$, we can find a (approximate) critical point by finding a (approximate) minimizer of $\mathcal{H}$. Moreover, under Assumption 2.6, finding a critical point is equivalent to finding a saddle point. This motivates the HGD update procedure on $x^{(k)} = (x_1^{(k)}, x_2^{(k)})$ with step-size $\eta > 0$:

$$x^{(k+1)} = x^{(k)} - \eta \nabla \mathcal{H}(x^{(k)}), \tag{2}$$

HGD has been mentioned in Mescheder et al. (2017); Balduzzi et al. (2018), and it strongly resembles the Consensus Optimization (CO) approach of Mescheder et al. (2017). The HGD update requires a Hessian-vector product because $\nabla \mathcal{H} = \xi^\top J$, making HGD a second-order iterative scheme. However, Hessian-vector products are cheap to compute when the objective is defined by a neural net, taking only two gradient oracle calls (Pearlmutter, 1994). This makes the Hessian-vector product oracle a theoretically appealing primitive, and it has been used widely in the nonconvex optimization literature. Since Hessian-vector product oracles are feasible to compute for GANs, many recent algorithms for local min-max nonconvex optimization have also utilized Hessian-vector products (Mescheder et al., 2017; Balduzzi et al., 2018; Adolphs et al., 2019; Letcher et al., 2019; Mazumdar et al., 2019).

---

[1]Specifically, they assume $g(x_1, x_2) = f(x_1) + x_2^T A x_1 - h(x_2)$, where $f$ is smooth and convex, $h$ is smooth and strongly convex, and $A$ has full column rank. We make a brief comparison of our work to that of Du & Hu (2019) for the convex-strongly concave setting in Appendix D.

[2]We note that the function $\mathcal{H}$ is not the Hamiltonian as in the sense of classical physics, as we do not use the symplectic structure in our analysis, but rather we only perform gradient descent on $\mathcal{H}$.

To the best of our knowledge, previous work on last-iterate convergence rates has only focused on how algorithms perform in three particular cases: (a) when the objective $g$ is bilinear, (b) when $g$ is strongly convex-strongly concave, and (c) when $g$ is convex-strongly concave (Tseng, 1995; Liang & Stokes, 2019; Du & Hu, 2019; Mokhtari et al., 2019). The existence of methods with provable finite-time guarantees for settings beyond the aforementioned has remained an open problem. This work is the first to show that an efficient algorithm, namely HGD, can achieve non-asymptotic convergence in settings that are not strongly convex or linear in either player.

## 3.1 Convergence Rates for HGD

We now state our main theorems for this paper, which show convergence to critical points. When Assumption 2.6 holds, we get convergence to min-maxes. All of our main results will use the following multi-part assumption:

**Assumption 3.1.** *Let $g : \mathbb{R}^d \times \mathbb{R}^d \to \mathbb{R}$.*

   1. *Assume a critical point for $g$ exists.*

   2. *Assume $g$ is $(L_1, L_2, L_3)$-Lipschitz and let $L_{\mathcal{H}} = L_1 L_3 + L_2^2$.*

Our first theorem shows that HGD converges for the strongly convex-strongly concave case. Although simple, this result will help us demonstrate our analysis techniques.

**Theorem 3.2.** *Let Assumption 3.1 hold and let $g(x_1, x_2)$ be $\alpha$-strongly convex in $x_1$ and $\alpha$-strongly concave in $x_2$. Then HGD with step-size $\eta = 1/L_{\mathcal{H}}$ starting from some $x^{(0)} \in \mathbb{R}^d \times \mathbb{R}^d$ will have the following convergence rate:* $\left|\left|\xi(x^{(k)})\right|\right| \le \left(1 - \frac{\alpha^2}{L_{\mathcal{H}}}\right)^{k/2} \left|\left|\xi(x^{(0)})\right|\right|.$

Next, we show that HGD converges when $g$ is linear in one of its arguments and the cross-derivative is full rank. This setting allows a slightly tighter analysis compared to Theorem 3.4.

**Theorem 3.3.** *Let Assumption 3.1 hold and let $g(x_1, x_2)$ be $L$-smooth in $x_1$ and linear in $x_2$, and assume the cross derivative $\nabla^2_{x_1, x_2} g$ is full rank with all singular values at least $\gamma > 0$ for all $x \in \mathbb{R}^d \times \mathbb{R}^d$. Then HGD with step-size $\eta = 1/L_{\mathcal{H}}$ starting from some $x^{(0)} \in \mathbb{R}^d \times \mathbb{R}^d$ will have the following convergence rate:* $\left|\left|\xi(x^{(k)})\right|\right| \le \left(1 - \frac{\gamma^4}{(2\gamma^2 + L^2)L_{\mathcal{H}}}\right)^{k/2} \left|\left|\xi(x^{(0)})\right|\right|.$

Finally, we show our main result, which requires smoothness in both players and a large, well-conditioned cross-derivative.

**Theorem 3.4.** *Let Assumption 3.1 hold and let $g$ be $L$-smooth in $x_1$ and $L$-smooth in $x_2$. Let $\mu^2 = \min_{x_1, x_2} \lambda_{\min}((\nabla^2_{x_2 x_2} g(x_1, x_2))^2)$ and $\rho^2 = \min_{x_1, x_2} \lambda_{\min}((\nabla^2_{x_1 x_1} g(x_1, x_2))^2)$, and assume the cross derivative $\nabla^2_{x_1 x_2} g$ is full rank with all singular values lower bounded by $\gamma > 0$ and upper bounded by $\Gamma$ for all $x \in \mathbb{R}^d \times \mathbb{R}^d$. Moreover, let the following "sufficiently bilinear" condition hold:*

$$(\gamma^2 + \rho^2)(\mu^2 + \gamma^2) - 4L^2\Gamma^2 > 0. \tag{3}$$

*Then HGD with step-size $\eta = 1/L_{\mathcal{H}}$ starting from some $x^{(0)} \in \mathbb{R}^d \times \mathbb{R}^d$ will satisfy*

$$\left|\left|\xi(x^{(k)})\right|\right| \le \left(1 - \frac{(\gamma^2 + \rho^2)(\gamma^2 + \mu^2) - 4L^2\Gamma^2}{(2\gamma^2 + \rho^2 + \mu^2)L_{\mathcal{H}}}\right)^{k/2} \left|\left|\xi(x^{(0)})\right|\right|. \tag{4}$$

As discussed above, Theorem 3.4 provides the first last-iterate convergence rate for min-max problems that does not require strong convexity or linearity in either input. For example, the objective $g(x_1, x_2) = f(x_1) + 3Lx_1^\top x_2 - h(x_2)$, where $f$ and $h$ are $L$-smooth convex functions, satisfies the assumptions of Theorem 3.4 and is not strongly convex or linear in either input. We discuss a simple example that is not convex-concave in Appendix E. We also show how our results can be applied to specific settings, such as the Dirac-GAN, in Appendix G.

The "sufficiently bilinear" condition (3) is in some sense necessary for our linear convergence rate since linear convergence is impossible in general for convex-concave settings, due to lower bounds on convex optimization (Agarwal & Hazan, 2018; Arjevani et al., 2017). We give some explanations for this condition in the following section. In simple experiments for HGD on convex-concave and nonconvex-nonconcave objectives, the convergence rate speeds up when there is a larger bilinear component, as expected from our theoretical results. We show these experiments in Appendix K.

## 3.2 EXPLANATION OF "SUFFICIENTLY BILINEAR" CONDITION

In this section, we explain the "sufficiently bilinear" condition (3). Suppose our objective is $g(x_1, x_2) = \hat{g}(x_1, x_2) + cx_1^\top x_2$ for a smooth function $\hat{g}$. Then for sufficiently large values of $c$ (i.e. $g$ has a large enough bilinear term), we see that $g$ satisfies (3). To see this, note that if we have $\gamma^4 > 4L^2\Gamma^2$, then condition (3) holds. Let $\gamma'$ and $\Gamma'$ be lower and upper bounds on the singular values of $\nabla^2_{x_1 x_2}\hat{g}$. Then it suffices to have $(\gamma' + c)^4 > 4L^2(\Gamma' + c)^2$, which is true for $c = 3\max\{L, \Gamma'\}$ (i.e. $c = O(L)$ suffices).

This condition is analogous to the case when we use SGDA on the objective $g(x_1, x_2) = \hat{g}(x_1, x_2) + c\,||x_1||^2 - c\,||x_2||^2$ for $L$-smooth convex-concave $\hat{g}$. According to Liang & Stokes (2019), SGDA will converge at a rate of roughly $\frac{\tilde{L}^2}{c^2}\log(1/\epsilon)$ for $\tilde{L}$-smooth and $c$-strongly convex-strongly concave objectives.[3] For $c = 0$, SGDA will diverge in the worst case. For $c = o(L)$, we get linear convergence, but it will be slow because $\frac{L+c}{c}$ is large (this can be thought of as a large condition number). Finally, for $c = \Omega(L)$, we get fast linear convergence, since $\frac{L+c}{c} = O(1)$. Thus, to get fast linear convergence it suffices to make the problem "sufficiently strongly convex-strongly concave" (or "sufficiently strongly monotone").

Theorem 3.4 and condition (3) show that there exists another class of settings where we can achieve linear rates in the min-max setting. In our case, if we have an objective $g(x_1, x_2) = \hat{g}(x_1, x_2) + cx_1^\top x_2$ for a smooth function $\hat{g}$, we will get linear convergence if $\|\nabla^2_{x_1 x_2}\hat{g}\| \leq \delta L$ and $c \geq 3(1+\delta)L$, which ensures that the problem is "sufficiently bilinear." Intuitively, it makes sense that the "sufficiently bilinear" setting allows a linear rate because the pure bilinear setting allows a linear rate.

Another way to understand condition (3) is that it is a sufficient condition for the existence of a unique critical point in a general class of settings, as we show in the following lemma, which we prove in Appendix F.

**Lemma 3.5.** *Let $g(x_1, x_2) = f(x_1) + cx_1^\top x_2 - h(x_2)$ where $f$ and $h$ are $L$-smooth. Moreover, assume that $\nabla^2 f(x_1)$ and $\nabla^2 h(x_2)$ each have a 0 eigenvalue for some $x_1$ and $x_2$. If (3) holds, then $g$ has a unique critical point.*

## 4 PROOF SKETCHES FOR HGD CONVERGENCE RATE RESULTS

In this section, we go over the key components of the proofs for our convergence rates from Section 3.1. Recall that the intuition behind HGD was that critical points (where $\xi(x) = 0$) are global minima of $\mathcal{H} = \frac{1}{2}||\xi||^2$. On the other hand, there is no guarantee that $\mathcal{H}$ is a convex potential function, and a priori, one would not assume gradient descent on this potential would find a critical point. Nonetheless, we are able to show that in a variety of settings, $\mathcal{H}$ satisfies the *PL condition*, which allows HGD to have linear convergence. Proving this requires proving properties about the singular values of $J \equiv \nabla\xi$.

### 4.1 THE POLYAK-ŁOJASIEWICZ CONDITION FOR THE HAMILTONIAN

We begin by recalling the definition of the PL condition.

**Definition 4.1** (Polyak-Łojasiewicz (PL) condition Polyak (1963); Lojasiewicz (1963)). *A function $f: \mathbb{R}^d \to \mathbb{R}$ satisfies the PL condition with parameter $\alpha > 0$ if for all $x \in \mathbb{R}^d$, $\frac{1}{2}||\nabla f(x)||^2 \geq \alpha(f(x) - \min_{x^* \in \mathbb{R}^d} f(x^*))$.*

The PL condition is well-known to be the weakest condition necessary to obtain linear convergence rate for gradient methods; see for example Karimi et al. (2016). We will show that $\mathcal{H}$ satisfies the PL condition, which allows us to use the following classic theorem.

**Theorem 4.2** (Linear rate under PL Polyak (1963); Lojasiewicz (1963)). *Let $f: \mathbb{R}^d \to \mathbb{R}$ be $L$-smooth and let $x^* \in \arg\min_{x \in \mathbb{R}^d} f(x)$. Suppose $f$ satisfies the PL condition with parameter $\alpha$. Then if we run gradient descent from $x^{(0)} \in \mathbb{R}^d$ with step-size $\frac{1}{L}$, we have: $f(x^{(k)}) - f(x^*) \leq (1 - \frac{\alpha}{L})^k(f(x^{(0)}) - f(x^*))$.*

---

[3]The actual rate is $\frac{\beta}{c}\log(1/\epsilon)$, for some parameter $\beta$ that is at least $(L + c)^2$.

For completeness, we provide the proof of Theorem 4.2 in Appendix C.

All of our results use Assumption 3.1, so we are guaranteed that $g$ has a critical point. This implies that the global minimum of $\mathcal{H}$ is 0, which allows us to prove the following key lemma:

**Lemma 4.3.** *Assume we have a twice differentiable $g(x_1, x_2)$ with associated $\xi, \mathcal{H}, J$. Let $c > 0$. If $JJ^\top \succeq \alpha I$ for every $x$, then $\mathcal{H}$ satisfies the PL condition with parameter $\alpha$.*

*Proof.* Consider the squared norm of the gradient of the Hamiltonian:

$$\tfrac{1}{2}\|\nabla\mathcal{H}\|^2 = \tfrac{1}{2}\|J^\top\xi\|^2 = \tfrac{1}{2}\langle\xi, (JJ^\top)\xi\rangle \geq \tfrac{\alpha}{2}\|\xi\|^2 = \alpha\mathcal{H}.$$

The proof is finished by noting that $\mathcal{H}(x) = 0$ when $x$ is a critical point. $\qquad\square$

To use Theorem 4.2, we will also need to show that $\mathcal{H}$ is smooth, which holds when $g$ is $(L_1, L_2, L_3)$-Lipschitz. The proof of Lemma 4.4 is in Appendix H.

**Lemma 4.4.** *Consider any $g(x_1, x_2)$ which is $(L_1, L_2, L_3)$-Lipschitz for constants $L_1, L_2, L_3 > 0$. Then the Hamiltonian $\mathcal{H}(x)$ is $(L_1 L_3 + L_2^2)$-smooth.*

To use Lemma 4.3, we will need control over the eigenvalues of $JJ^\top$, which we achieve with the following linear algebraic lemmas. We provide their proofs in Appendix H.

**Lemma 4.5.** *Let $H = \begin{pmatrix} M_1 & B \\ -B^\top & -M_2 \end{pmatrix}$ and let $\epsilon \geq 0$. If $M_1 \succ \epsilon I$ and $M_2 \prec -\epsilon I$, then for all eigenvalues $\lambda$ of $HH^\top$, we have $\lambda > \epsilon^2$.*

**Lemma 4.6.** *Let $H = \begin{pmatrix} A & C \\ -C^\top & 0 \end{pmatrix}$, where $C$ is square and full rank. Then if $\lambda$ is an eigenvalue of $HH^\top$, then we must have $\lambda \geq \frac{\sigma_{\min}^4(C)}{2\sigma_{\min}^2(C)+\|A\|^2}$.*

### 4.2 Proof sketches for Theorems 3.2, 3.3, and 3.4

We now proceed to sketch the proofs of our main theorems using the techniques we have described. The following lemma shows it suffices to prove the PL condition for $\mathcal{H}$ for the various settings of our theorems:

**Lemma 4.7.** *Given $g : \mathbb{R}^d \times \mathbb{R}^d \to \mathbb{R}$, suppose $\mathcal{H}$ satisfies the PL condition with parameter $\alpha$ and is $L_\mathcal{H}$-smooth. Then if we use HGD starting from some $x^{(0)} \in \mathbb{R}^d \times \mathbb{R}^d$ with step-size $\eta = 1/L_\mathcal{H}$, then we have the following:*

$$\left\|\xi(x^{(k)})\right\| \leq \left(1 - \tfrac{\alpha}{L_\mathcal{H}}\right)^{k/2} \left\|\xi(x^{(0)})\right\|.$$

*Proof.* Since $\mathcal{H}$ satisfies the PL condition with parameter $\alpha$ and $\mathcal{H}$ is $L_\mathcal{H}$-smooth, we know by Theorem 4.2 that gradient descent on $\mathcal{H}$ with step-size $1/L_\mathcal{H}$ converges at a rate of $\mathcal{H}(x^{(k)}) \leq (1 - \tfrac{\alpha}{L_\mathcal{H}})^k \mathcal{H}(x^{(0)})$. Substituting in for $\mathcal{H}$ gives the lemma. $\qquad\square$

It remains to show that $\mathcal{H}$ satisfies the PL condition in the settings of Theorems 3.2 to 3.4. First, we show the result for the strongly convex-strongly concave setting of Theorem 3.2.

**Lemma 4.8** (PL for the strongly convex-strongly concave setting). *Let $g$ be $c$-strongly convex in $x_1$ and $c$-strongly concave in $x_2$. Then $\mathcal{H}$ satisfies the PL condition with parameter $\alpha = c^2$.*

*Proof.* We apply Lemma 4.5 with $H = J$. Since $g$ is $c$-strongly-convex in $x_1$ and $c$-strongly concave in $x_2$ we have $M_1 = \nabla^2_{x_1 x_1} g \succ cI$ and $M_2 = -\nabla^2_{x_2 x_2} g \succ cI$. Then the magnitude of the eigenvalues of $J$ is at least $c$. Thus, $JJ^\top \succeq c^2 I$, so by Lemma 4.3, $\mathcal{H}$ satisfies the PL condition with parameter $c^2$. $\qquad\square$

Next, we show that $\mathcal{H}$ satisfies the PL condition for the nonconvex-linear setting of Theorem 3.3. We prove this lemma in Appendix H.4 by using Lemma 4.6.

**Lemma 4.9** (PL for the smooth nonconvex-linear setting). *Let $g$ be $L$-smooth in $x_1$ and linear in $x_2$. Moreover, for all $x \in \mathbb{R}^d \times \mathbb{R}^d$, let $\nabla^2_{x_1 x_2} g(x_1, x_2)$ be full rank and square with $\sigma_{\min}(\nabla^2_{x_1 x_2} g(x_1, x_2)) \geq \gamma$. Then $\mathcal{H}$ satisfies the PL condition with parameter $\alpha = \frac{\gamma^4}{2\gamma^2 + L^2}$.*

Finally, we prove that $\mathcal{H}$ satisfies the PL condition in the nonconvex-nonconvex setting of Theorem 3.4. The proof for Lemma 4.10 is in Appendix H.5, and it uses Lemma H.2, which is similar to Lemma 4.6.

**Lemma 4.10** (PL for the smooth nonconvex-nonconvex setting). *Let $g$ be $L$-smooth in $x_1$ and $L$-smooth in $x_2$. Also, let $\nabla^2_{x_1 x_2} g$ be full rank and let all of its singular values be lower bounded by $\gamma$ and upper bounded by $\Gamma$ for all $x \in \mathbb{R}^d \times \mathbb{R}^d$. Let $\rho^2 = \min_{x_1, x_2} \lambda_{\min}((\nabla^2_{x_1 x_1} g(x_1, x_2))^2)$ and $\mu^2 = \min_{x_1, x_2} \lambda_{\min}((\nabla^2_{x_2 x_2} g(x_1, x_2))^2)$. Assume the following condition holds:*

$$(\gamma^2 + \rho^2)(\gamma^2 + \mu^2) - 4L^2 \Gamma^2 > 0.$$

*Then $\mathcal{H}$ satisfies the PL condition with parameter $\alpha = \frac{(\gamma^2 + \rho^2)(\gamma^2 + \mu^2) - 4L^2 \Gamma^2}{2\gamma^2 + \rho^2 + \mu^2}$.*

Combining Lemmas 4.8 to 4.10 with Lemma 4.7 yields Theorems 3.2 to 3.4.

## 5 EXTENSIONS OF HGD RESULTS

**Stochastic HGD** Our results above also imply rates for stochastic HGD, where the gradient $\nabla \mathcal{H}$ in (2), is replaced by a stochastic estimator $v$ of $\nabla \mathcal{H}$ such that $\mathrm{E}[v] = \nabla \mathcal{H}$. Since we show that $\mathcal{H}$ satisfies the PL condition with parameter $\alpha$ in different settings, we can use Theorem 4 in Karimi et al. (2016) to show that stochastic HGD converges at a $O(1/\sqrt{k})$ rate in the settings of Theorems 3.2 to 3.4, including the "sufficiently bilinear" setting. We prove Theorem 5.1 in Appendix I.

**Theorem 5.1.** *Let Assumption 3.1 hold and suppose $\mathcal{H}$ satisfies the PL condition with parameter $\alpha$. Suppose we use the update $x^{(k+1)} = x^{(k)} - \eta_k v(x^{(k)})$, where $v$ is a stochastic estimate of $\nabla \mathcal{H}$ such that $\mathrm{E}[v] = \nabla \mathcal{H}$ and $\mathrm{E}[\|v(x^{(k)})\|^2] \leq C^2$ for all $x^{(k)}$. Then if we use $\eta_k = \frac{2k+1}{2\alpha(k+1)^2}$, we have the following convergence rate: $\mathrm{E}[\|\xi(x^{(k)})\|] \leq \sqrt{\frac{L_{\mathcal{H}} C^2}{k \alpha^2}}$.*

**Consensus Optimization** The Consensus Optimization (CO) algorithm of Mescheder et al. (2017) is as follows:

$$x^{(k+1)} = x^{(k)} - \eta(\xi(x^{(k)}) + \gamma \nabla \mathcal{H}(x^{(k)})) \tag{5}$$

where $\gamma > 0$. This is essentially a weighted combination of SGDA and HGD. Mescheder et al. (2017) remark that while HGD has poor performance on nonconvex problems in practice, CO can effectively train GANs in a variety of settings, including on CIFAR-10 and celebA. While they frame CO as SGDA with a small modification, they actually set $\gamma = 10$ for several of their experiments, which suggests that one can also view CO as a modified form of HGD.

Using this perspective, we prove Theorem 5.2, which implies that we get linear convergence of CO in the same settings as Theorems 3.2 to 3.4 provided that $\gamma$ is sufficiently large (i.e. the HGD update is large compared to the SGDA update). The key technical component is showing that HGD still performs well even with a certain kind of small arbitrary perturbation. Previously, Liang & Stokes (2019) proved that CO achieves linear convergence in the bilinear setting, so our result greatly expands the settings where CO has provable non-asymptotic convergence. We prove Theorem 5.2 in Appendix J.

**Theorem 5.2.** *Let Assumption 3.1 hold. Let $g$ be $L_g$ smooth and suppose $\mathcal{H}$ satisfies the PL condition with parameter $\alpha$. Then if we update some $x^{(0)} \in \mathbb{R}^d \times \mathbb{R}^d$ using the CO update (5) with step-size $\eta = \frac{\alpha}{4 L_{\mathcal{H}} L_g}$ and $\gamma = \frac{4 L_g}{\alpha}$, we get the following convergence:*

$$\left\| \xi(x^{(k)}) \right\| \leq \left(1 - \frac{\alpha}{4 L_{\mathcal{H}}}\right)^k \left\| \xi(x^{(0)}) \right\|. \tag{6}$$

We also show that CO converges in practice on some simple examples in Appendix K.

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

# A    GENERAL NONCONVEX MIN-MAX OPTIMIZATION

In standard nonconvex optimization, a common goal is to find second-order local minima, which are approximate critical points where $\nabla^2 f$ is approximately positive definite. Likewise, a common goal in nonconvex min-max optimization is to find approximate critical points where an analogous second-order condition holds, namely that $\nabla^2_{x_1 x_1} g(x)$ is approximately positive definite and $\nabla^2_{x_2 x_2} g(x)$ is approximately negative definite. Critical points where this second-order condition holds are called *local min-maxes*. When Assumption 2.6 holds, all critical points are *global* min-maxes, but in more general settings, we may encounter critical points that do not satisfy these conditions. Critical points may be local min-mins or max-mins or indefinite points. A number of recent papers have proposed dynamics for nonconvex min-max optimization, showing local stability or local asymptotic convergence results (Mescheder et al., 2017; Daskalakis & Panageas, 2018; Balduzzi et al., 2018; Letcher et al., 2019; Mazumdar et al., 2019). The key guarantee that these papers generally give is that their algorithms will be stable at local min-maxes and unstable at some set of undesirable critical points (such as local max-mins). This essentially amounts to a guarantee that in the convex-concave setting, their algorithms will converge asymptotically and in the strictly concave-strictly convex setting (i.e. where there is only an undesirable *max-min*), their algorithms will diverge asymptotically. This type of local stability is essentially the best one can ask for in the general nonconvex setting, and we show how to give similar guarantees for our algorithm in Section A.1.

## A.1    NONCONVEX EXTENSIONS FOR HGD

While the naive version of HGD will try to converge to all critical points, we can modify HGD slightly to achieve second-order stability guarantees as in various related work such as Balduzzi et al. (2018); Letcher et al. (2019). In particular, we consider modifying HGD so that there is some scalar $\alpha$ in front of the $\nabla \mathcal{H}$ term as follows:

$$x^{(k+1)} = x^{(k)} - \eta \alpha \nabla \mathcal{H}(x^{(k)}) \tag{7}$$

We now present two ways to choose $\alpha$. Our first method is inspired by the Simplectic Gradient Adjustment algorithm of Balduzzi et al. (2018), which is as follows:

$$x^{(k+1)} = x^{(k)} - \eta (\xi(x^{(k)}) - \lambda A^\top \xi(x^k)) \tag{8}$$

where $A$ is the antisymmetric part of $J$ and $\lambda = \mathrm{sgn}\left(\langle \xi, J \rangle \left\langle A^\top \xi, J \right\rangle\right)$. Balduzzi et al. (2018) show that $\lambda$ is positive when in a strictly convex-strictly concave region and negative in a strictly concave-strictly convex region. Thus, if we choose $\alpha = \lambda = \mathrm{sgn}\left(\langle \xi, J \rangle \left\langle A^\top \xi, J \right\rangle\right)$, we can ensure that the modified HGD will exhibit local stability around strict min-maxes and local instability around strict max-mins. This follows simply because we will do gradient *descent* on $\mathcal{H}$ in the first case and gradient *ascent* on $\mathcal{H}$ in the second case.

Another way to choose $\alpha$ involves using an approximate eigenvalue computation on $\nabla^2_{x_1 x_1} g$ and $\nabla^2_{x_2 x_2} g$ to detect whether $\nabla^2_{x_1 x_1} g$ is positive semidefinite and $\nabla^2_{x_2 x_2} g$ is negative semidefinite (which would mean we are in a convex-concave region). We set $\alpha = 1$ if we are in a convex-concave region and $-1$ otherwise, which will guarantee local stability around min-maxes and local instability around other critical points. This approximate eigenvector computation can be done using a logarithmic number of Hessian-vector products.

# B    BACKGROUND ON NON-UNIFORM AVERAGE ITERATES

A number of recent works have focused on the performance of a non-uniform average of an algorithm's iterates. Iterate averaging can lend stability to an algorithm or improve performance if the algorithm cycles around the solution. On the other hand, uniform averages can suffer from worse performance in nonconvex settings if early iterates are far from optimal. Non-uniform averaging is a way to achieve the stability benefits of iterate averaging while potentially speeding up convergence compared to uniform averaging. In this way, one can view non-uniform averaging as an interpolation between average-iterate and last-iterate algorithms.

One popular non-uniform averaging scheme is the exponential moving average (EMA). For an algorithm with iterates $z^{(0)}, ..., z^{(T)}$, the EMA at iterate $t$ is defined recursively as

$$z_{EMA}^{(t)} = \beta z_{EMA}^{(t-1)} + (1 - \beta) z_{EMA}^{(t-1)}$$

where $z_{EMA}^{(0)} = z^{(0)}$ and $\beta < 1$. A typical value for $\beta$ is 0.999. Yazıcı et al. (2019) and Gidel et al. (2019) show that uniform and EMA schemes can improve GAN performance on a variety of datasets. Mescheder et al. (2018) and Karras et al. (2018) use EMA to evaluate the GAN models they train, showing the effectiveness of EMA in practice.

In terms of theoretical results, Kroer (2019) studies saddle point problems of the form $\min_{x_1} \max_{x_2} f(x_1) + g(x_1) + \langle Kx_1, x_2 \rangle - h^*(x_2)$, where $f$ is a smooth convex function, $g$ and $h$ are convex functions with easily computable prox-mappings, and $K$ is some linear operator. They show that for certain algorithms, linear averaging and quadratic averaging schemes are provably at least as good as the uniform average scheme in terms of iterate complexity. Abernethy et al. (2018) show how linear and exponential averaging schemes can be used to achieve faster convergence rates in some specific convex-concave games.

Overall, while non-uniform averaging is appealing for a variety of reasons, there is currently no theoretical explanation for why it outperforms uniform averages or why it would converge at all in many settings. In fact, one natural way to show convergence for an EMA scheme would be to show last-iterate convergence.

## C PROOF OF LINEAR CONVERGENCE RATE UNDER PL CONDITION

Here we present a classic proof of Theorem 4.2.

*Proof of Theorem 4.2.*

$$f(x^{(k+1)}) - f(x^*) \leq f(x^{(k)}) - f(x^*) - \frac{1}{2L} \left\| \nabla f(x^{(k)}) \right\|^2 \tag{9}$$

$$\leq f(x^{(k)}) - f(x^*) - \frac{\alpha}{L}(f(x^{(k)}) - f(x^*)) \tag{10}$$

$$= \left(1 - \frac{\alpha}{L}\right)(f(x^{(k)}) - f(x^*)) \tag{11}$$

where the first line comes from smoothness and the update rule for gradient descent, the second inequality comes from the PL condition. Applying the last line recursively gives the result. $\square$

## D COMPARISON OF THEOREM 3.4 TO DU & HU (2019)

In this section, we compare our results in Theorem 3.4 to those of Du & Hu (2019). Du & Hu (2019) prove a rate for SGDA when $g$ is $L$-smooth and convex in $x_1$ and $L$-smooth and $\mu$-strongly concave in $x_2$ and $\nabla_{x_1 x_2}^2 g$ is some fixed matrix $A$. The specific setting they consider is to find the unconstrained min-max for a function $g : \mathbb{R}^{d_1} \times \mathbb{R}^{d_2} \to \mathbb{R}$ defined as $g(x_1, x_2) = f(x_1) + x_2^\top A x_1 - h(x_2)$ where $f$ is convex and smooth, $h$ is strongly-convex and smooth, and $A \in \mathbb{R}^{d_2 \times d_1}$ has rank $d_1$ (i.e. $A$ has full column rank).

Their rate uses the potential function $P_t = \lambda a_t + b_t$, where we have:

$$\lambda = \frac{2L\Gamma(L + \frac{\Gamma^2}{\mu})}{\mu \gamma^2} \tag{12}$$

$$a_k = \left\| x_1^{(k)} - x_1^* \right\| \tag{13}$$

$$b_k = \left\| x_2^{(k)} - x_2^* \right\| \tag{14}$$

where $(x_1^*, x_2^*)$ is the min-max for the objective. Their rate (Theorem 3.1 in Du & Hu (2019)) is

$$P_{k+1} \leq \left(1 - c\frac{\mu^2 \gamma^4}{L^3 \Gamma^2 (L + \frac{\Gamma^2}{\mu})}\right)^k P_k \tag{15}$$

for some constant $c > 0$. To translate this rate into bounds on $||\xi||$, we can use the smoothness of $g$ in both of its arguments to note that $\left|\left|\frac{\partial g}{\partial x_1}(x_1, x_2)\right|\right| = \left|\left|\frac{\partial g}{\partial x_1}(x_1, x_2) - \frac{\partial g}{\partial x_1}(x_1^*, x_2^*)\right|\right| \leq L\left|\left|x_1^{(k)} - x_1^*\right|\right|$ and likewise for $x_2$. So the rate on $P_k$ translates into a rate on $||\xi||$ with some additional factor in front.

Their rate and our rate are incomparable – neither is strictly better. For instance when $\gamma = \Gamma$ is much larger than all other quantities, their rates simplify to $\left(1 - O\left(\frac{\mu^3}{L^3}\right)\right)^k$, while ours go to $\left(1 - O\left(\frac{\gamma^2}{L\mathcal{H}}\right)\right)^{k/2}$. While our convergence rate requires the sufficiently bilinear condition (3) to hold, we do not require convexity in $x_1$ or concavity in $x_2$. Moreover, we allow $\nabla^2_{x_1 x_2} g$ to change as long as the bounds on the singular values hold whereas Du & Hu (2019) require $\nabla^2_{x_1 x_2} g$ to be a fixed matrix.

## E  NONCONVEX-NONCONCAVE SETTING WHERE ASSUMPTION 2.6 AND THE CONDITIONS FOR THEOREM 3.4 HOLD

In this section we give a concrete example of a nonconvex-nonconcave setting where Assumption 2.6 and the conditions for Theorem 3.4 hold. We choose this example for simplicity, but one can easily come up with other more complicated examples.

For our example, we define the following function:

$$F(x) = \begin{cases} -3(x + \frac{\pi}{2}) & \text{for } x \leq -\frac{\pi}{2} \\ -3\cos x & \text{for } -\frac{\pi}{2} < x \leq \frac{\pi}{2} \\ -\cos x + 2x - \pi & \text{for } x > \frac{\pi}{2} \end{cases} \tag{16}$$

The first and second derivatives of $F$ are as follows:

$$F'(x) = \begin{cases} -3 & \text{for } x \leq -\frac{\pi}{2} \\ 3\sin x & \text{for } -\frac{\pi}{2} < x \leq \frac{\pi}{2} \\ \sin x + 2 & \text{for } x > \frac{\pi}{2} \end{cases} \tag{17}$$

$$F''(x) = \begin{cases} 0 & \text{for } x \leq -\frac{\pi}{2} \\ 3\cos x & \text{for } -\frac{\pi}{2} < x \leq \frac{\pi}{2} \\ \cos x & \text{for } x > \frac{\pi}{2} \end{cases} \tag{18}$$

From Figure 2, we can see that this function is neither convex nor concave.

Our objective will be $g(x_1, g_2) = F(x_1) + 4x_1^\top x_2 - F(x_2)$. Note that $L = 3$ because $F''(x) \leq 3$ for all $x$. Also, $\gamma = \Gamma = 4$ since $\nabla^2_{x_1 x_2} g = 4I$.

First, we show that $g$ satisfies Assumption 3.1. We see that $g$ has a critical point at $(0, 0)$. Moreover, $g$ is $(L_1, L_2, L_3)$-Lipschitz for any finite-sized region of $\mathbb{R}^2$. Thus, if we assume our algorithm stays within a ball of some radius $R$, the $(L_1, L_2, L_3)$-Lipschitz assumption will be satisfied. Since our algorithm does not diverge and indeed converges at a linear rate to the min-max, this assumption is fairly mild.

Next, we show that $g$ satisfies condition (3). Condition (3) requires $\gamma^4 > 4L^2\Gamma^2$ for $g$. We see that this holds because $\gamma^4 = 4^4 = 256$ and $4L\Gamma^2 = 4 * 3 * 4^2 = 192$.

Therefore, the assumptions of Theorem 3.4 are satisfied.

We can also show that this objective satisfies Assumption 2.6, so we get convergence to the min-max of $g$. We will show that $g$ has only one critical point (at $(0, 0)$) and that this critical point is a min-max. We first give a "proof by picture" below, showing a plot of $g$ in Figure 3, along with plots of $g(\cdot, 0)$ and $g(0, \cdot)$ showing that $(0, 0)$ is indeed a min-max.

We can also formally show that $(0, 0)$ is the unique critical point of $g$ and that it is a min-max. We prove this for completeness, although the calculations more or less amount to a simple case analysis.

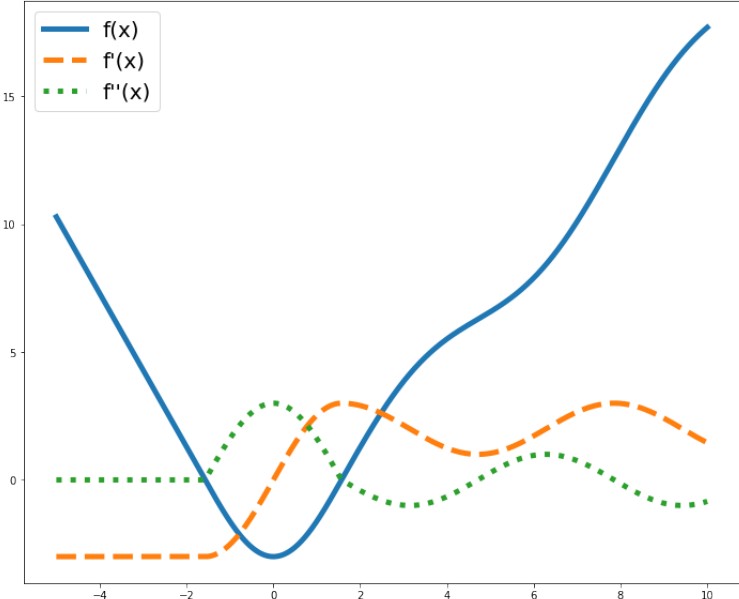

Figure 2: Plot of nonconvex function $F(x)$ defined in (16), as well as its first and second derivatives

Let us look at the derivatives of $g$ with respect to $x_1$ and $x_2$:

$$\frac{\partial g}{\partial x_1}(x_1, x_2) = \begin{cases} -3 + 4x_2 & \text{for } x_1 \leq -\frac{\pi}{2} \\ 3\sin x_1 + 4x_2 & \text{for } -\frac{\pi}{2} < x_1 \leq \frac{\pi}{2} \\ \sin x_1 + 2 + 4x_2 & \text{for } x_1 > \frac{\pi}{2} \end{cases} \tag{19}$$

$$\frac{\partial g}{\partial x_2}(x_1, x_2) = \begin{cases} 3 + 4x_1 & \text{for } x_2 \leq -\frac{\pi}{2} \\ -3\sin x_2 + 4x_1 & \text{for } -\frac{\pi}{2} < x_2 \leq \frac{\pi}{2} \\ -\sin x_2 + 2 + 4x_1 & \text{for } x_2 > \frac{\pi}{2} \end{cases} \tag{20}$$

Observe that if $x_1 \in [-\frac{\pi}{2}, \frac{\pi}{2}]$ then critical points of $g$ must satisfy $3\sin x_1 + 4x_2 = 0$, which implies that $x_2 \in [-\frac{3}{4}, \frac{3}{4}]$. Likewise, if $x_2 \in [-\frac{\pi}{2}, \frac{\pi}{2}]$, then critical points of $g$ must have $x_1 \in [-\frac{3}{4}, \frac{3}{4}]$. We show that this implies that $g$ only has critical points where $x_1$ and $x_2$ are both in the range $[-\frac{\pi}{2}, \frac{\pi}{2}]$.

Suppose $g$ had a critical point such that $x_1 \leq -\frac{\pi}{2}$. Then this critical point must satisfy $x_2 = \frac{3}{4}$. But from our observation above, if a critical point has $x_2 = \frac{3}{4}$, then $x_1$ must lie in $[-\frac{3}{4}, \frac{3}{4}]$, which contradicts $x_1 \leq -\frac{\pi}{2}$.

Next, suppose $g$ had a critical point such that $x_1 > \frac{\pi}{2}$. Then this critical point must satisfy $x_2 = -\frac{1}{4}(\sin x_1 + 2)$, which implies that $x_2 \in [-\frac{3}{4}, \frac{3}{4}]$. But then by the observation above, $x_1$ must lie in $[-\frac{3}{4}, \frac{3}{4}]$, which contradicts $x_1 > \frac{\pi}{2}$.

From this we see that any critical point of $g$ must have $x_1 \in [-\frac{\pi}{2}, \frac{\pi}{2}]$. We can make analogous arguments to show that any critical point of $g$ must have $x_2 \in [-\frac{\pi}{2}, \frac{\pi}{2}]$.

From this, we can conclude that all critical points of $g$ must satisfy the following:

$$3\sin x_1 + 4x_2 = 0 \tag{21}$$
$$-3\sin x_2 + 4x_1 = 0 \tag{22}$$

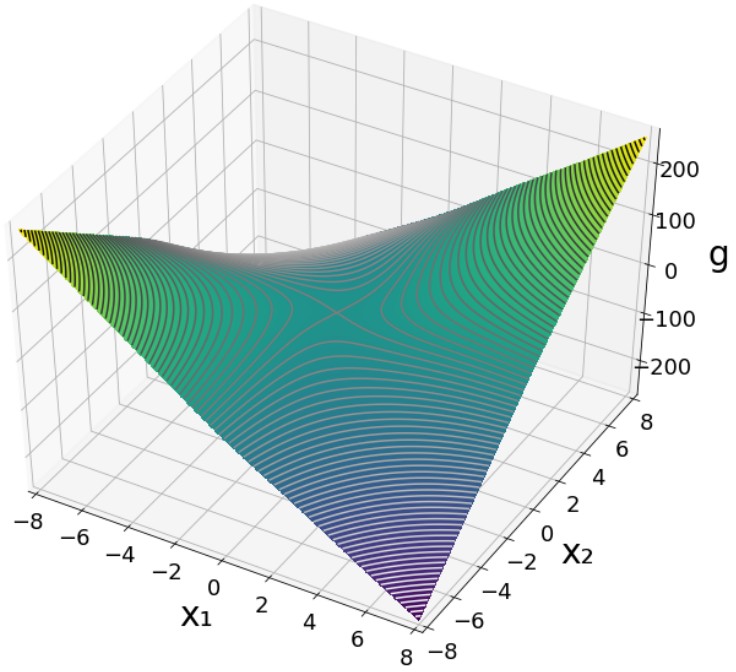

Figure 3: Plot of nonconvex-nonconcave $g(x_1, x_2) = F(x_1) + 4x_1^\top x_2 - F(x_2)$

These equations imply the following:

$$x_1 = \frac{3}{4} \sin x_2 \tag{23}$$

$$x_2 = -\frac{3}{4} \sin x_1 \tag{24}$$

$$\Rightarrow x_1 = \frac{3}{4} \sin \left( -\frac{3}{4} \sin x_2 \right) \tag{25}$$

$$\Rightarrow x_2 = -\frac{3}{4} \sin \left( \frac{3}{4} \sin x_2 \right) \tag{26}$$

That is, for all critical points of $g$, $x_1$ must be a fixed point of $h_1(x) = \frac{3}{4} \sin \left( -\frac{3}{4} \sin x \right)$ and $x_2$ must be a fixed point of $h_2(x) = -\frac{3}{4} \sin \left( \frac{3}{4} \sin x \right)$. Since $|h_1'(x)| < 1$ and $|h_2'(x)| < 1$ always, $h_1$ and $h_2$ are contractive maps, so they have only one fixed point each. Thus, $g$ will only have one critical point, namely the point $(x_1, x_2)$ such that $x_1$ is the unique fixed point of $h_1$ and $x_2$ is the unique fixed point of $h_2$.

Finally, we can observe that $(0, 0)$ is a critical point of $g$, so it must be the unique critical point of $g$. One can also see that this is a min-max by looking at the second derivatives of $F$ in (18).

## F    PROOF OF LEMMA 3.5

To prove Lemma 3.5, we will use the following lemma:

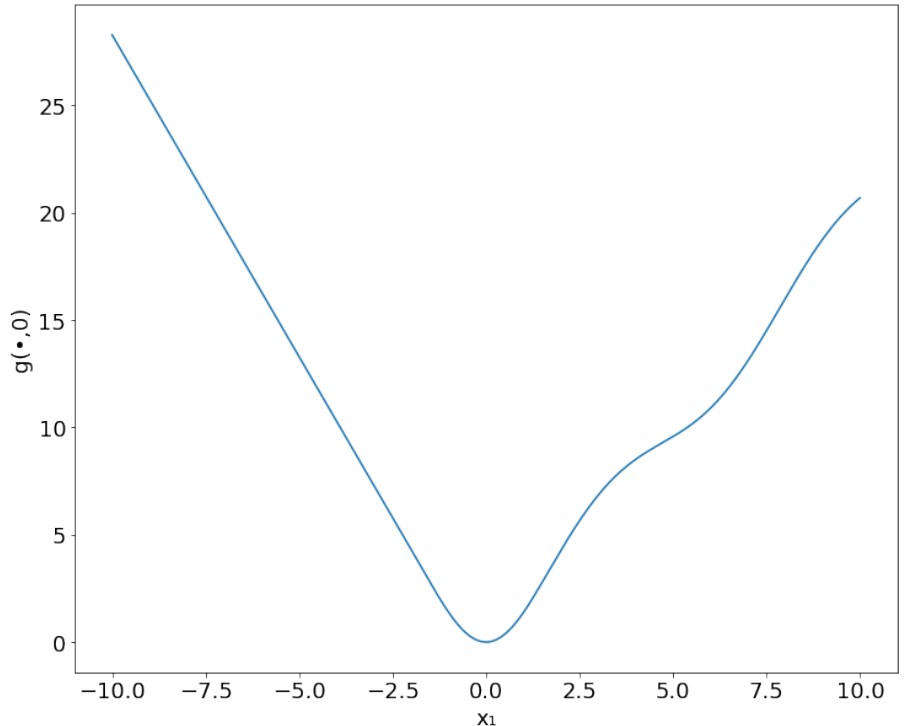

Figure 4: Plot of $g(\cdot, 0)$. We can see that there is only one min and it occurs at $x_1 = 0$.

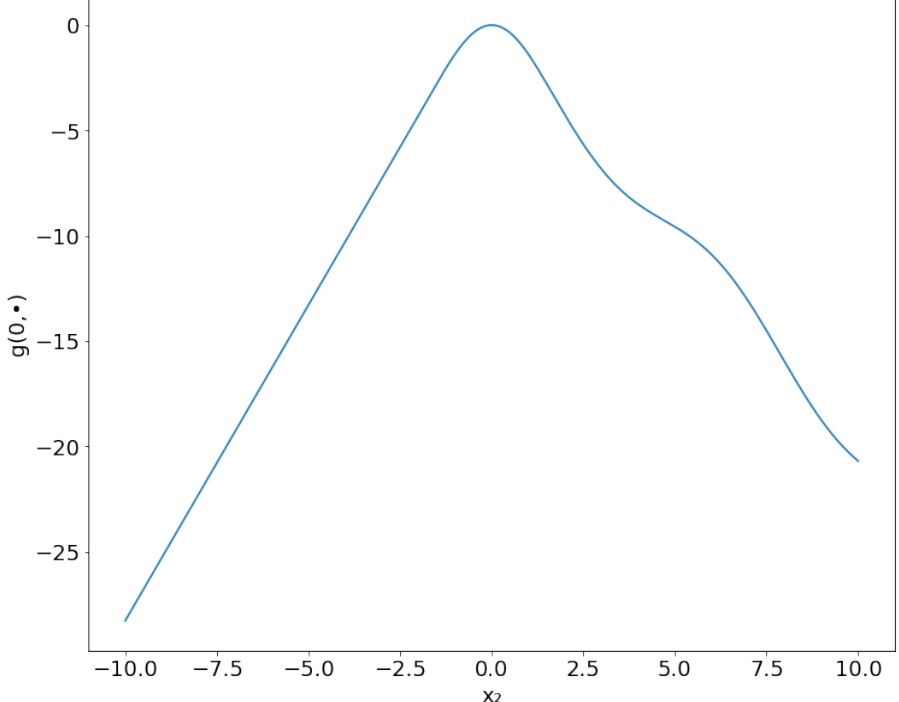

Figure 5: Plot of $g(0, x_2)$. We can see that there is only one max and it occurs at $x_2 = 0$.

**Lemma F.1.** *Let $g(x_1, x_2) = f(x_1) + cx_1^\top x_2 - h(x_2)$ where $f$ and $h$ are $L$-smooth. Then if $c > L$, $g$ has a unique critical point.*

*Proof of Lemma 3.5.* Condition (3) is as follows:

$$(\gamma^2 + \rho^2)(\mu^2 + \gamma^2) - 4L^2\Gamma^2 > 0. \tag{27}$$

Note that in our setting, $\gamma = \Gamma = c$. Next, observe that if $\nabla^2 f(x_1)$ and $\nabla^2 h(x_2)$ each have a 0 eigenvalue for some $x_1$ and $x_2$, condition (3) reduces to:

$$c > 2L. \tag{28}$$

Then by Lemma F.1, we see that $g$ must have a unique critical point. □

Next, we prove Lemma F.1.

*Proof of Lemma F.1.* Suppose our objective is $g(x_1, x_2) = f(x_1) + cx_1^\top x_2 - h(x_2)$ where $f$ and $h$ are both $L$-smooth convex functions. Critical points of $g$ must satisfy the following:

$$\nabla f(x_1) + cx_2 = 0 \tag{29}$$

$$-\nabla h(x_2) + cx_1 = 0 \tag{30}$$

$$\Rightarrow x_1 = \frac{1}{c}\nabla h(x_2) \tag{31}$$

$$\Rightarrow x_2 = -\frac{1}{c}\nabla f\left(\frac{1}{c}\nabla h(x_2)\right) \tag{32}$$

In other words, $x_2$ must be a fixed point of $F(z) = -\frac{1}{c}\nabla f(\frac{1}{c}\nabla h(z))$. The function $F$ will have a unique fixed point if it is a contractive map. We now show that for $c > L$, this is the case.

$$||F(u) - F(v)|| = \left|\left|\frac{1}{c}\nabla f\left(\frac{1}{c}\nabla h(u)\right) - \frac{1}{c}\nabla f\left(\frac{1}{c}\nabla h(v)\right)\right|\right| \tag{33}$$

$$\leq \frac{L}{c} \cdot \left|\left|\frac{1}{c}\nabla h(u) - \frac{1}{c}\nabla h(v)\right|\right| \tag{34}$$

$$\leq \frac{L^2}{c^2}||u - v|| < ||u - v|| \tag{35}$$

where the inequalities follow from smoothness of $f$ and $h$. An analogous property can be shown by solving for $x_1$ instead. Thus, if $c > L$, then $g$ will have a unique fixed point.

Condition (3) is thus a sufficient condition for the existence of a unique critical point for the class of objectives above. □

## G  APPLICATIONS

In this section, we discuss how our results can be applied to various settings. One simple setting is the Dirac-GAN from Mescheder et al. (2018), where $g(x_1, x_2) = \min_{x_1}\max_{x_2} f(x_1^\top x_2) - f(0)$ for some function $f$ whose derivative is always non-zero. When $f(t) = t$, the Dirac-GAN is just a bilinear game, so HGD will converge globally to the Nash Equilibrium (NE) of this Dirac-GAN, as shown in Balduzzi et al. (2018). Our results prove global convergence rates for HGD on the Dirac-GAN even when a small smooth convex regularizer is added for the discriminator or subtracted for the generator. Moreover, Lemma 2.2 of Mescheder et al. (2018) shows that the diagonal blocks of the Jacobian are 0 at the NE for arbitrary $f$ with non-zero derivative. As such, HGD will achieve the convergence rates in this paper in a region around the NE for the Dirac-GAN for arbitrary $f$ with non-zero derivative even when a small smooth convex regularizer is added for either player.

Du & Hu (2019) list several applications where the min-max formulation is relevant, such as in ERM problems with a linear classifier. Given a data matrix $A$, the ERM problem involves solving $\min_x \ell(Ax) + f(x)$ for some smooth, convex loss $\ell$ and smooth, convex regularizer $f$. This problem has the saddle point formulation $\min_x \max_y y^\top Ax - \ell^*(y) + f(x)$. According to Du & Hu (2019), this formulation can be advantageous when it allows a finite-sum structure, reduces communication complexity in a distributed setting, or allows some sparsity structure to be exploited. Our results show that linear rates are possible for this problem if $A$ is square, well-conditioned, and sufficiently large compared to $\ell$ and $f$.

# H    PROOFS FOR SECTION 4

In this section, we prove our main results about the convergence of HGD, starting with some key technical lemmas.

## H.1    PROOF OF LEMMA 4.4

*Proof.* We have $\nabla \mathcal{H} = \xi^\top J$. Let $u, v \in \mathbb{R}^d \times \mathbb{R}^d$. Then we have:

$$
\begin{aligned}
||\nabla \mathcal{H}(u) - \nabla \mathcal{H}(v)|| &= \left\|\xi^\top(u) J(u) - \xi^\top(v) J(v)\right\| \\
&= \left\|\xi^\top(u) J(u) - \xi^\top(u) J(v) + \xi^\top(u) J(v) - \xi^\top(v) J(v)\right\| \\
&\leq \left\|\xi^\top(u) J(u) - \xi^\top(u) J(v)\right\| + \left\|\xi^\top(u) J(v) - \xi^\top(v) J(v)\right\| \\
&\leq ||\xi(u)|| \cdot ||J(u) - J(v)|| + ||\xi(u) - \xi(v)|| \cdot ||J(v)|| \\
&\leq (L_1 L_3 + L_2^2) \, ||u - v|| \qquad\qquad\qquad\qquad \square
\end{aligned}
$$

## H.2    PROOF OF LEMMA 4.5

*Proof.* Note that $HH^\top = \begin{pmatrix} M_1^2 + BB^T & -M_1 B - BM_2 \\ -(M_1 B + BM_2)^T & M_2^2 + B^T B \end{pmatrix} = \begin{pmatrix} M_1 & -B \\ -B^T & M_2 \end{pmatrix}^2$.

Now let $Z = \begin{pmatrix} M_1 & -B \\ -B^T & M_2 \end{pmatrix}$. It suffices to show that for any eigenvalue $\delta$ of $Z$, $|\delta| \leq \epsilon$. For the sake of contradiction, let $v$ be an eigenvalue of $Z$ with eigenvalue $\delta$ such that $|\delta| \leq \epsilon$. Let $v = \begin{pmatrix} v_1 \\ v_2 \end{pmatrix}$. Since $Zv = \delta v$ for $|\delta| \leq \epsilon$ and $M_1 \succ \epsilon I$ and $M_2 \prec -\epsilon I$, we must have $v_1 \neq 0$ and $v_2 \neq 0$. Then we have:

$$
\begin{pmatrix} M_1 v_1 - B v_2 \\ M_2 v_2 - B^\top v_1 \end{pmatrix} = \delta \begin{pmatrix} v_1 \\ v_2 \end{pmatrix} \tag{36}
$$

This implies

$$
(M_1 - \delta I) v_1 = B v_2 \tag{37}
$$

$$
(M_2 - \delta I) v_2 = B^\top v_1 \tag{38}
$$

Let $\hat{M}_1 = M_1 - \delta I$ and let $\hat{M}_2 = M_2 - \delta I$. Note that $\hat{M}_1 \succ 0$ and $\hat{M}_2 \prec 0$. Then we can write $v_1 = \hat{M}_1^{-1} B v_2$. Further, we can substitute into (38) to get

$$
\hat{M}_2 v_2 = B^\top \hat{M}_1^{-1} B v_2 \tag{39}
$$

$$
\Longleftrightarrow \quad -\hat{M}_2^{-1} B^\top \hat{M}_1^{-1} B v_2 = -v_2 \tag{40}
$$

In other words, $v_2$ is an eigenvector of $-\hat{M}_2^{-1} B^\top \hat{M}_1^{-1} B$ with eigenvalue $-1$. Let $A = -\hat{M}_2^{-1}$ and $T = B^\top \hat{M}_1^{-1} B$. Note that $A$ is positive definite and $T$ is PSD. Then we have:

$$
AT = A^{1/2}(A^{1/2} T A^{1/2}) A^{-1/2} \tag{41}
$$

Since $A^{1/2} T A^{1/2}$ is PSD, and $AT$ is similar to $A^{1/2} T A^{1/2}$, we must have that all of the eigenvalues of $AT$ are nonnegative. This contradicts that $v_2$ is an eigenvector of $AT$ with eigenvalue $-1$.

Thus, all eigenvalues of $Z$ must have magnitude greater than $\epsilon$. $\qquad\square$

## H.3    PROOF OF LEMMA 4.6

*Proof.* Suppose $\lambda$ is an eigenvalue of $HH^\top$ with eigenvector $v = \begin{pmatrix} v_1 \\ v_2 \end{pmatrix}$. WLOG, suppose $\lambda < \sigma_{\min}^2(C)$. Since $v$ is an eigenvector, we have:

$$
\begin{pmatrix} A^2 + CC^\top & -AC \\ -C^\top A & C^\top C \end{pmatrix} \begin{pmatrix} v_1 \\ v_2 \end{pmatrix} = \lambda \begin{pmatrix} v_1 \\ v_2 \end{pmatrix} \tag{42}
$$

Thus, we have:

$$(A^2 + CC^\top - \lambda I)v_1 - ACv_2 = 0 \tag{43}$$

$$-C^\top A v_1 + (C^\top C - \lambda I)v_2 = 0 \tag{44}$$

Since $\lambda < \sigma_{\min}^2(C)$, we have that $C^\top C - \lambda I$ is invertible, so we can write $v_2 = (C^\top C - \lambda I)^{-1}C^\top A v_1$ from the (44). Plugging this into (43) gives:

$$(A^2 + CC^\top - \lambda I - AC(C^\top C - \lambda I)^{-1}C^\top A)v_1 = 0 \tag{45}$$

$$(A(I - C(C^\top C - \lambda I)^{-1}C^\top)A + CC^\top - \lambda I)v_1 = 0 \tag{46}$$

Write the SVD of $C$ as $C = U\Sigma V^\top$. Then we have:

$$C(C^\top C - \lambda I)^{-1}C^\top = U\Sigma V^\top (V\Sigma U^\top U\Sigma V^\top - \lambda I)^{-1}V\Sigma U^\top \tag{47}$$

$$= U\Sigma V^\top (V(\Sigma^2 - \lambda I)V^\top)^{-1}V\Sigma U^\top \tag{48}$$

$$= U\Sigma V^\top V^{-T}(\Sigma^2 - \lambda I)^{-1}V^{-1}V\Sigma U^\top \tag{49}$$

$$= U\Sigma^2(\Sigma^2 - \lambda I)^{-1}U^\top \tag{50}$$

$$= UDU^\top \tag{51}$$

where the second line follows because $VV^\top = I$ when $C$ is full rank and where $D$ is a diagonal matrix such that $D_{ii} = \frac{\sigma_i^2(C)}{\sigma_i^2(C) - \lambda}$.

Let $M = I - D$, so $M$ is diagonal with $M_{ii} = \frac{-\lambda}{\sigma_i^2(C) - \lambda}$. Then (46) becomes:

$$(AMA + CC^\top - \lambda I)v_1 = 0 \tag{52}$$

This means $T = AMA + CC^\top - \lambda I$ has a 0 eigenvalue. A simple lower bound for the eigenvalues of $T$ is

$$\lambda_{\min}(T) \geq -||A||^2 \frac{\lambda}{\sigma_{\min}^2 - \lambda} + \sigma_{\min}^2(C) - \lambda \tag{53}$$

We will show that if $\lambda < \delta$, where $\delta = \sigma_{\min}^2(C) + \frac{||A||^2}{2} - \sqrt{(\sigma_{\min}^2 + \frac{||A||^2}{2})^2 - \sigma_{\min}^4}$, then $\lambda_{\min}(T) > 0$, which is a contradiction. It suffices to show the following inequality:

$$-||A||^2 \frac{\lambda}{\sigma_{\min}^2 - \lambda} + \sigma_{\min}^2(C) - \lambda > 0 \tag{54}$$

$$\iff \sigma_{\min}^2(C) - \lambda > ||A||^2 \frac{\lambda}{\sigma_{\min}^2 - \lambda} \tag{55}$$

$$\iff (\sigma_{\min}^2(C) - \lambda)^2 > ||A||^2 \lambda \tag{56}$$

$$\iff \lambda^2 - (2\sigma_{\min}^2(C) + ||A||^2)\lambda + \sigma_{\min}^4(C) > 0 \tag{57}$$

(57) has zeros at the following values:

$$\sigma_{\min}^2(C) + \frac{||A||^2}{2} \pm \sqrt{\left(\sigma_{\min}^2 + \frac{||A||^2}{2}\right)^2 - \sigma_{\min}^4(C)} \tag{58}$$

Since (57) is a convex parabola, if $\lambda$ is less than both zeros, we will have proved (57). This is clearly true if $\lambda < \delta$.

As a last step, we can give a slightly nicer form of $\delta$, using Lemma H.1. Letting $x = \sigma_{\min}^2(C) + \frac{||A||^2}{2}$ and $c = \sigma_{\min}^4(C)$, we have $\delta > \frac{\sigma_{\min}^4(C)}{2\sigma_{\min}^2(C) + ||A||^2}$. So to reiterate, if $\lambda < \frac{\sigma_{\min}^4(C)}{2\sigma_{\min}^2(C) + ||A||^2} < \delta$, then (57) holds, so $T \succ 0$, which contradicts (52). $\qquad\square$

**Lemma H.1.** *For $x \in (0, 1)$ and $c \in (0, x^2)$, we have:*

$$x - \sqrt{x^2 - c} > \frac{c}{2x}$$

*Proof.*

$$x - \sqrt{x^2 - c} = x - x\sqrt{1 - \frac{c}{x^2}} > x - x\left(1 - \frac{c}{2x^2}\right) = \frac{c}{2x}$$

$\square$

### H.4 PROOF OF LEMMA 4.9

*Proof.* Let $C(x_1, x_2) = \nabla^2_{x_1 x_2} g(x_1, x_2)$. For all $x \in \mathbb{R}^d \times \mathbb{R}^d$, $C(x_1, x_2)$ is square and full rank by assumption, so we can apply Lemma 4.6 with $H = J$ at each point $x \in \mathbb{R}^d \times \mathbb{R}^d$, which gives $\lambda(JJ^\top) \geq \frac{\sigma^4_{\min}(C(x_1, x_2))}{2\sigma^2_{\min}(C(x_1, x_2)) + \left\|\nabla^2_{x_1 x_1} g(x_1, x_2)\right\|^2}$. We have $\left\|\nabla^2_{x_1 x_1} g(x_1, x_2)\right\| \leq L$ since $g$ is smooth in $x_1$. Also, $\sigma^2_{\min}(C(x_1, x_2)) \geq \gamma$. Then we have that $JJ^\top \succeq \frac{\gamma^4}{2\gamma^2 + L^2} I$, so by Lemma 4.3, $\mathcal{H}$ satisfies the PL condition with parameter $\frac{\gamma^4}{2\gamma^2 + L^2}$. $\square$

### H.5 PROOF OF LEMMA 4.10

To prove Lemma 4.10, we use the following lemma:

**Lemma H.2.** *Let* $H = \begin{pmatrix} A & C \\ -C^\top & -B \end{pmatrix}$, *where $C$ is square and full rank. Moreover, let* $c = (\sigma^2_{\min}(C) + \lambda_{\min}(A^2))(\lambda_{\min}(B^2) + \sigma^2_{\min}(C)) - \sigma^2_{\max}(C)(\|A\| + \|B\|)^2$ *and assume $c > 0$. Then if $\lambda$ is an eigenvalue of* $HH^\top = \begin{pmatrix} A^2 + CC^\top & -AC - CB \\ -C^\top A - BC^\top & B^2 + C^\top C \end{pmatrix}$, *we must have*

$$\lambda \geq \frac{(\sigma^2_{\min}(C) + \lambda_{\min}(A^2))(\lambda_{\min}(B^2) + \sigma^2_{\min}(C)) - \sigma^2_{\max}(C)(\|A\| + \|B\|)^2}{(2\sigma^2_{\min}(C) + \lambda_{\min}(A^2) + \lambda_{\min}(B^2))^2}.$$

*Proof of Lemma H.2.* This proof resembles that of Lemma 4.6. Let $v = \begin{pmatrix} v_1 \\ v_2 \end{pmatrix}$ be an eigenvector of $HH^\top$ with eigenvalue $\lambda$. Expanding $HH^\top v = \lambda v$, we have:

$$(A^2 + CC^\top - \lambda I)v_1 - (AC + CB)v_2 = 0 \tag{59}$$

$$-(C^\top A + BC^\top)v_1 + \underbrace{(B^2 + C^\top C - \lambda I)}_{M} v_2 = 0 \tag{60}$$

$$\Rightarrow v_2 = M^{-1}(C^\top A + BC^\top)v_1 \tag{61}$$

$$\Rightarrow (-(AC + CB)M^{-1}(C^\top A + BC^\top) + A^2 + CC^\top - \lambda I)v_1 = 0 \tag{62}$$

where $M$ is invertible because $C^\top C$ is positive definite and WLOG, we may assume that $\lambda < \lambda_{\min}(C^\top C) = \sigma^2_{\min}(C)$. We will show that if the assumptions in the statement of the lemma hold, then we get a contradiction if $\lambda$ is below some positive threshold. In particular, we show that the following inequality holds for small enough $\lambda$ (this inequality contradicts (62)):

$$\sigma^2_{\min}(C) - \lambda + \lambda_{\min}(A^2) > \sigma^2_{\max}(C)(\|A\| + \|B\|)^2 \left\|M^{-1}\right\|$$

$$\Leftarrow \sigma^2_{\min}(C) - \lambda + \lambda_{\min}(A^2) > \frac{\sigma^2_{\max}(C)}{\lambda_{\min}(B^2) + \sigma^2_{\min}(C) - \lambda}(\|A\| + \|B\|)^2$$

$$\Longleftrightarrow \lambda^2 - (2\sigma^2_{\min}(C) + \lambda_{\min}(A^2) + \lambda_{\min}(B^2))\lambda +$$

$$(\sigma^2_{\min}(C) + \lambda_{\min}(A^2))(\lambda_{\min}(B^2) + \sigma^2_{\min}(C)) - \sigma^2_{\max}(C)(\|A\| + \|B\|)^2 > 0$$

Letting $b = 2\sigma^2_{\min}(C) + \lambda_{\min}(A^2) + \lambda_{\min}(B^2)$, we can solve for the zeros of the above equation:

$$\lambda = \frac{b \pm \sqrt{b^2 - 4c}}{2} \tag{63}$$

Note that we have $c > 0$ by assumption, so this equation has only positive roots. Note also that $b^2 > 4c$, so the roots will not be imaginary. Then we see that if $\lambda < \delta = \frac{b - \sqrt{b^2 - 4c}}{2}$, we get

a contradiction. Using Lemma H.1, we see that $\delta > \frac{c}{b}$. So we've proven that $\lambda < \frac{c}{b}$ gives a contradiction, so we must have $\lambda \geq \frac{c}{b}$, i.e.

$$\lambda \geq \frac{(\sigma_{\min}^2(C) + \lambda_{\min}(A^2))(\lambda_{\min}(B^2) + \sigma_{\min}^2(C)) - \sigma_{\max}^2(C)(\|A\| + \|B\|)^2}{2\sigma_{\min}^2(C) + \lambda_{\min}(A^2) + \lambda_{\min}(B^2)}.$$

$\square$

*Proof of Lemma 4.10.* The proof is very similar to that of Lemma 4.9. Let $C(x_1, x_2) = \nabla_{x_1 x_2}^2 g(x_1, x_2)$. For all $x \in \mathbb{R}^d \times \mathbb{R}^d$, $C(x_1, x_2)$ is square and full rank with bounds on its singular values by assumption. Moreover, (3) holds, so we can apply Lemma H.2 with $H = J$ at each point $x \in \mathbb{R}^d \times \mathbb{R}^d$. Using the fact that $g$ is smooth in $x_1$ and $x_2$, this gives

$$\lambda(JJ^\top) \geq \frac{(\sigma_{\min}^2(C(x_1, x_2)) + \lambda_{\min}(A^2))(\sigma_{\min}^2(C(x_1, x_2)) + \mu^2) - 4L^2\sigma_{\max}^2(C(x_1, x_2))}{2\sigma_{\min}^2(C(x_1, x_2)) + \lambda_{\min}(A^2) + \mu^2}.$$

Using the bounds on the singular values of $C(x_1, x_2)$, we have that $JJ^\top \succeq \frac{(\gamma^2 + \lambda_{\min}(A^2))(\gamma^2 + \mu^2) - 4L^2\Gamma^2}{2\gamma^2 + \lambda_{\min}(A^2) + \mu^2} I$, so by Lemma 4.3, $\mathcal{H}$ satisfies the PL condition with parameter $\frac{(\gamma^2 + \lambda_{\min}(A^2))(\gamma^2 + \mu^2) - 4L^2\Gamma^2}{2\gamma^2 + \lambda_{\min}(A^2) + \mu^2}$.

$\square$

## I    PROOF OF THEOREM 5.1

In this section, we prove Theorem 5.1. The proof leverages the following theorem from Karimi et al. (2016).[4]

**Theorem I.1** (Karimi et al. (2016)). *Assume that $f$ is $L$-smooth, has a non-empty solution set $\mathcal{X}^*$, and satisfies the PL condition with parameter $\alpha$. Let $v$ be a stochastic estimate of $\nabla f$ such that $E[v] = \nabla f$. Assume $E[\|v(x^{(k)})\|^2] \leq C^2$ for all $x^{(k)}$ and some $C$. If we use the SGD update $x^{(k+1)} = x^{(k)} - \eta_k v(x^{(k)})$ with $\eta_k = \frac{2k+1}{2\alpha(k+1)^2}$, then, we get a convergence rate of*

$$E[f(x_k) - f^*] \leq \frac{LC^2}{2k\alpha^2} \tag{64}$$

*If instead we use a constant $\eta_k = \eta < \frac{1}{2\alpha}$, then we obtain a linear convergence rate up to a solution level that is proportional to $\eta$,*

$$E[f(x^{(k)}) - f^*] \leq (1 - 2\alpha\eta)^k [f(x^{(0)}) - f^*] + \frac{LC^2\eta}{4\alpha} \tag{65}$$

Now we can prove Theorem 5.1.

*Proof of Theorem 5.1.* If $\mathcal{H}$ satisfies the PL condition with parameter $\alpha$, then we can apply Theorem I.1 to the stochastic variant of HGD. since $\mathcal{H}^* = 0$, we get

$$E\left[\frac{1}{2}\|\xi(x^{(k)})\|^2\right] \leq \frac{L_{\mathcal{H}}C^2}{2k\alpha^2} \tag{66}$$

The theorem follows from Jensen's inequality, which implies that $E\left[\|\xi(x^{(k)})\|\right] \leq \sqrt{E\left[\|\xi(x^{(k)})\|^2\right]}$.

$\square$

## J    PROOF OF THEOREM 5.2

In this section, we prove our main result about Consensus Optimization, namely Theorem 5.2. The key technical component is showing that HGD still performs well even with small arbitrary perturbations, as we show in the following theorem:

---

[4]The actual theorem in Karimi et al. (2016) is stated in a slightly different way, but it is equivalent to our presentation.

**Theorem J.1.** *Let $x^{(k+1)} = x^{(k)} - \eta\nabla\mathcal{H}(x^{(k)}) + \eta_v v^{(k)}$ where $v^{(k)}$ is some arbitrary vector such that $\left|\left|v^{(k)}\right|\right| = \left|\left|\xi(x^{(k)})\right|\right|$. Let $g$ be $L_g$-smooth and suppose $\mathcal{H}$ satisfies the PL condition with parameter $\alpha$. Let $\eta = \frac{1}{L_\mathcal{H}}$ and let $\eta_v = \frac{\alpha}{4L_\mathcal{H}L_g}$. Then we get the following convergence:*

$$\left|\left|\xi(x^{(k)})\right|\right| \leq \left(1 - \tfrac{\alpha}{4L_\mathcal{H}}\right)^k \left|\left|\xi(x^{(0)})\right|\right|. \tag{67}$$

From Theorem J.1, it is simple to prove Theorem 5.2

*Proof of Theorem 5.2.* Note that the CO update (5) with $\gamma = \frac{4L_g}{\alpha}$ is exactly the update in Theorem J.1 with $v^{(k)} = -\xi(x^{(k)})$, so we get the desired convergence rate. □

Our result treats SGDA as an adversarial perturbation even though this is not the case, which suggests that this analysis may be improved. It would be nice if one could directly apply the PL-based analysis that we used for HGD, but this does not seem to work for CO since CO is not gradient descent on some objective.

Now we prove Theorem J.1.

*Proof of Theorem J.1.* Let $x^{(k+1/2)} = x^{(k)} - \eta\nabla\mathcal{H}(x^{(k)})$, so $x^{(k+1)} = x^{(k+1/2)} + \eta_v v^{(k)}$. From (11) in the proof of Theorem 4.2 with $\eta = \frac{1}{L_\mathcal{H}}$, we get

$$\left|\left|\xi(x^{(k+1/2)})\right|\right| \leq \left(1 - \tfrac{\alpha}{L_\mathcal{H}}\right)^{1/2} \left|\left|\xi(x^{(k)})\right|\right| \leq (1 - \tfrac{\alpha}{2L_\mathcal{H}}) \left|\left|\xi(x^{(k)})\right|\right|. \tag{68}$$

Next, note that the triangle inequality and smoothness of $g$ imply:

$$\left|\left|\xi(x^{(k+1)})\right|\right| \leq \left|\left|\xi(x^{(k+1/2)})\right|\right| + \left|\left|\xi(x^{(k+1)}) - \xi(x^{(k+1/2)})\right|\right| \tag{69}$$

$$\leq \left|\left|\xi(x^{(k+1/2)})\right|\right| + L_g \left|\left|x^{(k+1)} - x^{(k+1/2)}\right|\right| \tag{70}$$

$$= \left|\left|\xi(x^{(k+1/2)})\right|\right| + L_g \left|\left|\eta_v v\right|\right| \tag{71}$$

Using the above result and $\left|\left|v^{(k)}\right|\right| = \left|\left|\xi(x^{(k)})\right|\right|$, we get:

$$\left|\left|\xi(x^{(k+1)})\right|\right| \leq \left(1 - \frac{\alpha}{2L_\mathcal{H}} + L_g\eta_v\right) \left|\left|\xi(x^{(k)})\right|\right| \tag{72}$$

Setting $\eta_v = \frac{\alpha}{4L_\mathcal{H}L_g}$ gives the result. □

Note that for this result, we assume $g$ is $L_g$ smooth in $x_1$ and $x_2$ jointly, whereas in other parts of the paper we assume $g$ is smooth in $x_1$ or $x_2$ separately. If $g$ is $L$-smooth in $x_1$ and $L$-smooth in $x_2$ and $\left|\left|\nabla^2_{x_1 x_2} g(x_1, x_2)\right|\right| \leq L_c$ for all $x_1, x_2$, then $g$ will be $L + L_c$ smooth.

## K  EXPERIMENTS

In this section, we present some experimental results showing how SGDA, HGD, and CO perform on a convex-concave objective and a nonconvex-nonconcave objective. For our CO plots, $\gamma$ refers to the $\gamma$ parameter in the CO algorithm. All of our experiments are initialized at $(5, 5)$. The step-size $\eta$ for HGD and SGDA is always 0.01, while the step-size $\eta$ for CO with $\gamma = \{0.1, 1, 10\}$ is $\{0.1, 0.01, 0.001\}$ respectively to account for the fact that increasing $\gamma$ increases the effective step-size, so the $\eta$ parameter needs to be decreased accordingly. The experiments were all run on a standard 2017 Macbook Pro.

The main takeaways from the experiments are that CO with low $\gamma$ will not converge if there is a large bilinear term, while CO with high $\gamma$ and HGD all converge for small and large bilinear terms. When the bilinear term is large, CO with high $\gamma$ and HGD both will converge in fewer iterations (for the same step-size). We did not optimize for step-size, so it is possible this effect may change if the optimal step-size is chosen for each setting.

## K.1 CONVEX-CONCAVE OBJECTIVE

The convex-concave objective we use is $g(x_1, x_2) = f(x_1) + cx_1x_2 - f(x_2)$ where $f(x) = \log(1 + e^x)$. We show a plot of $f$ in Figure 6.

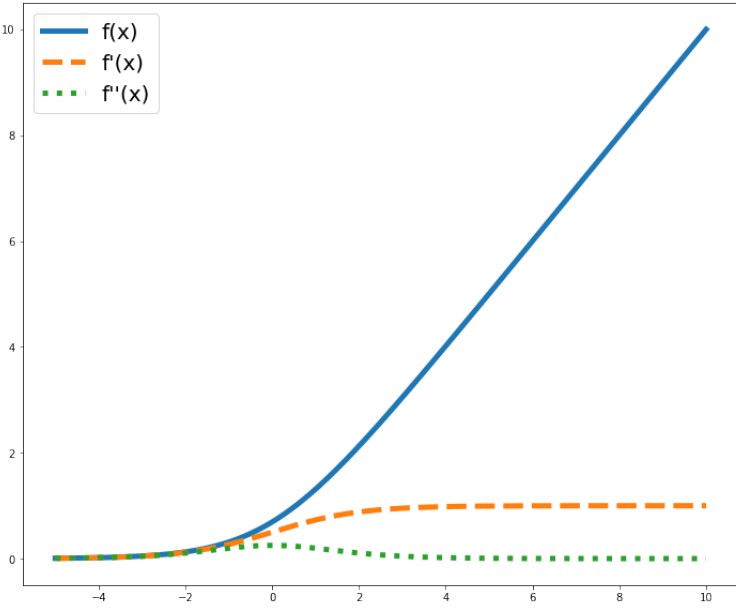

Figure 6: Plot of $f(x) = \log(1 + e^x)$ with its first and second derivatives. This is a convex, smooth function

When $c = 3$, SGDA converges, and when $c = 10$, SGDA diverges. We note that HGD and CO (for large enough $\gamma$) tend to converge faster when $c$ is larger.

### K.1.1 SGDA CONVERGES ($c = 3$)

These plots show $g$ when $c = 3$, so SGDA converges, as does CO with $\gamma = 0.1$.

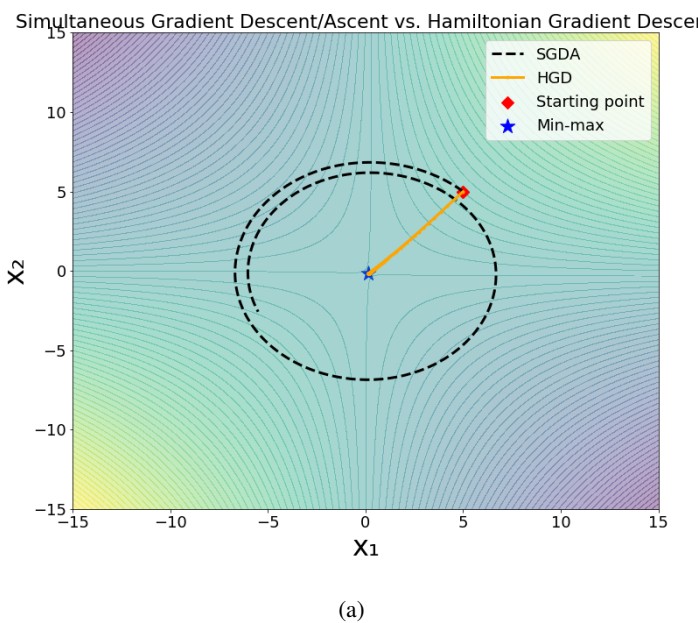

(a)

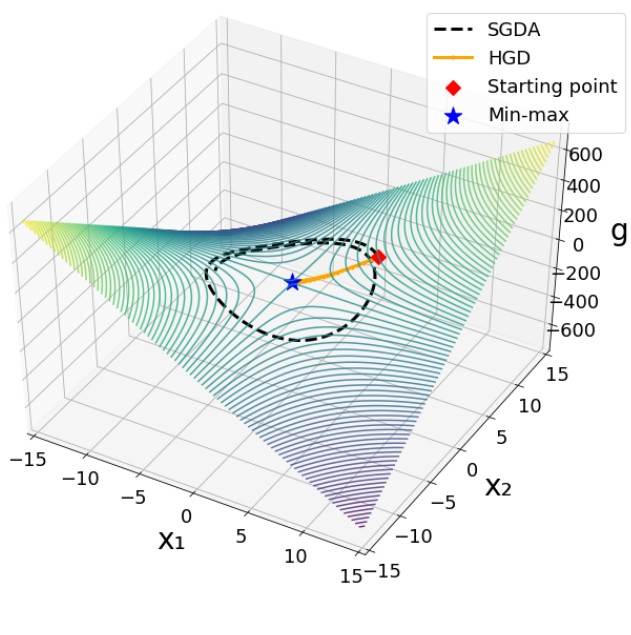

(b)

Figure 7: SGDA vs. HGD for 300 iterations for $g(x_1, x_2) = f(x_1) + cx_1x_2 - f(x_2)$ where $f(x) = \log(1 + e^x)$ and $c = 3$. SGDA slowly circles towards the min-max, and HGD goes directly to the min-max.

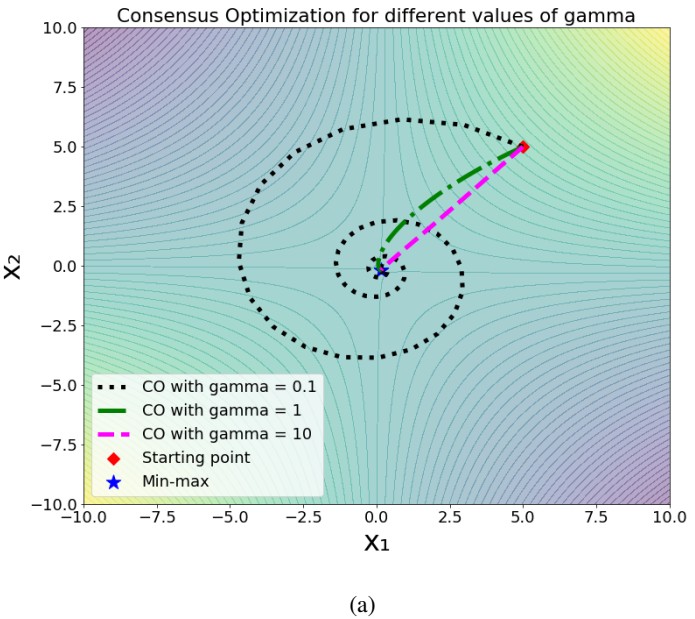

(a)

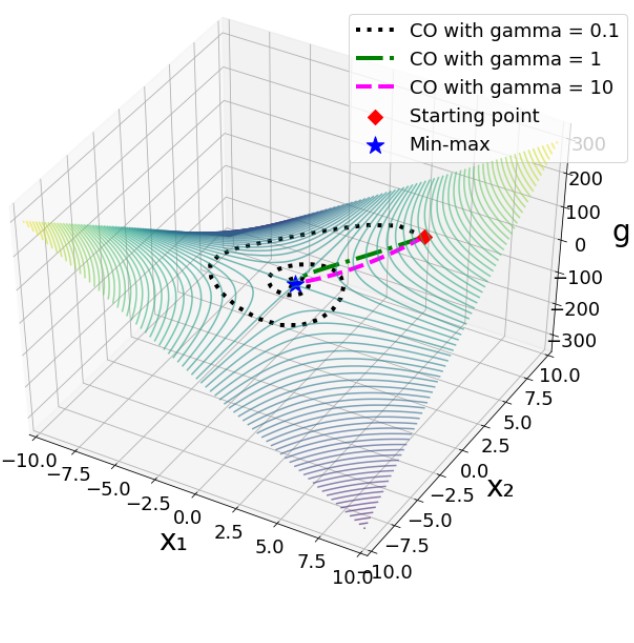

(b)

Figure 8: CO for 100 iterations with different values of $\gamma$ for $g(x_1, x_2) = f(x_1) + cx_1x_2 - f(x_2)$ where $f(x) = \log(1 + e^x)$ and $c = 3$. The $\gamma = 0.1$ curve slowly circles towards the min-max, while the other curves go directly to the min-max.

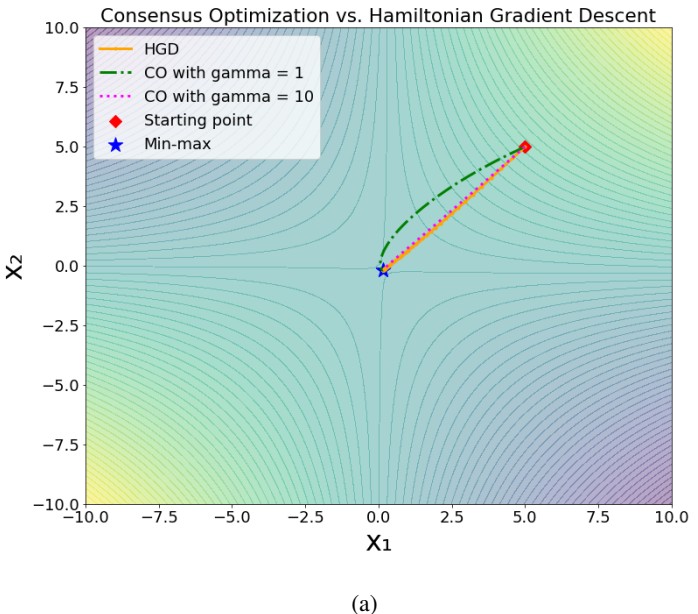

(a)

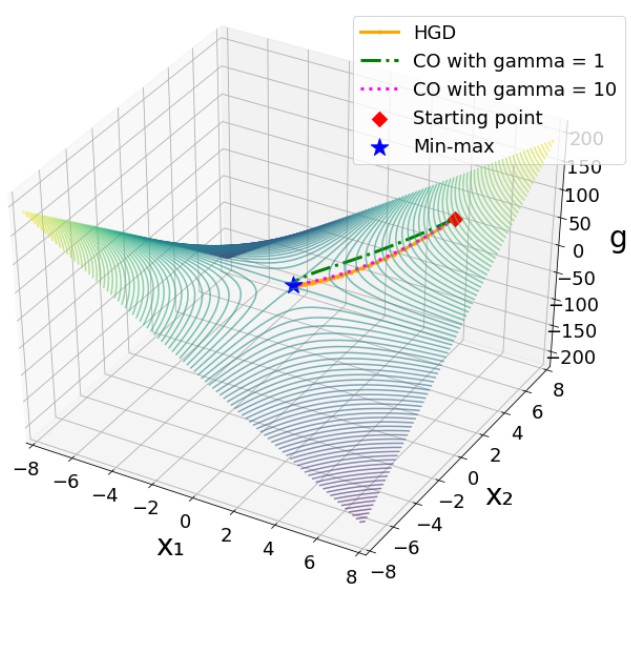

(b)

Figure 9: HGD vs. CO for 100 iterations for $g(x_1, x_2) = f(x_1) + cx_1 x_2 - f(x_2)$ where $f(x) = \log(1 + e^x)$ and $c = 3$ with different values of $\gamma$.

### K.1.2 SGDA DIVERGES ($c = 10$)

These plots show $g$ when $c = 10$, so SGDA diverges, as does CO with $\gamma = 0.1$. Note that in this case, CO with $\gamma \geq 1$ and HGD both require very few iterations (typically about 2) to reach the min-max.

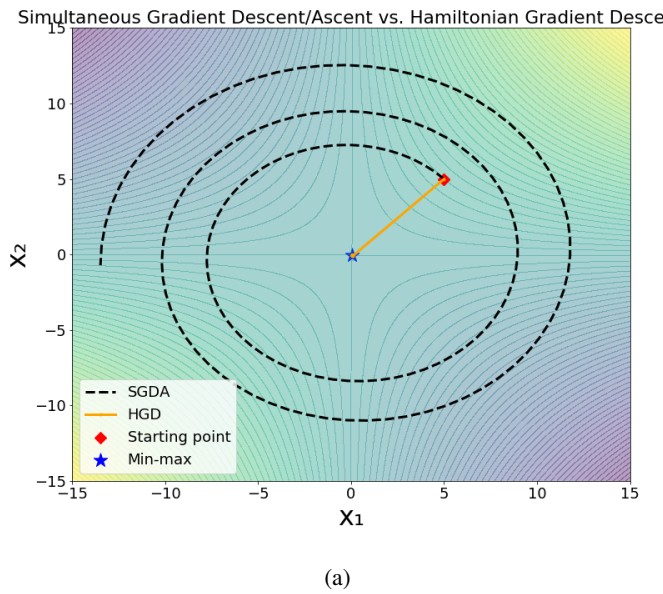

(a)

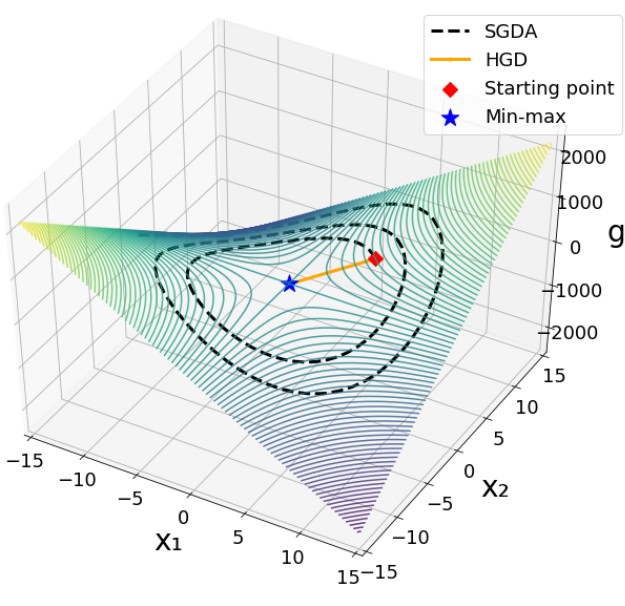

(b)

Figure 10: SGDA vs. HGD for 150 iterations for $g(x_1, x_2) = f(x_1) + cx_1x_2 - f(x_2)$ where $f(x) = \log(1 + e^x)$ and $c = 10$. SGDA slowly circles away from the min-max, while HGD goes directly to the min-max.

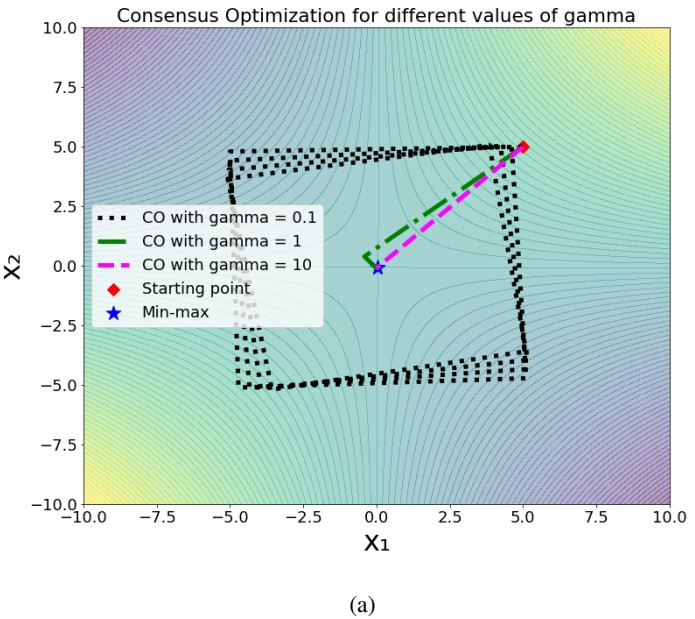

(a)

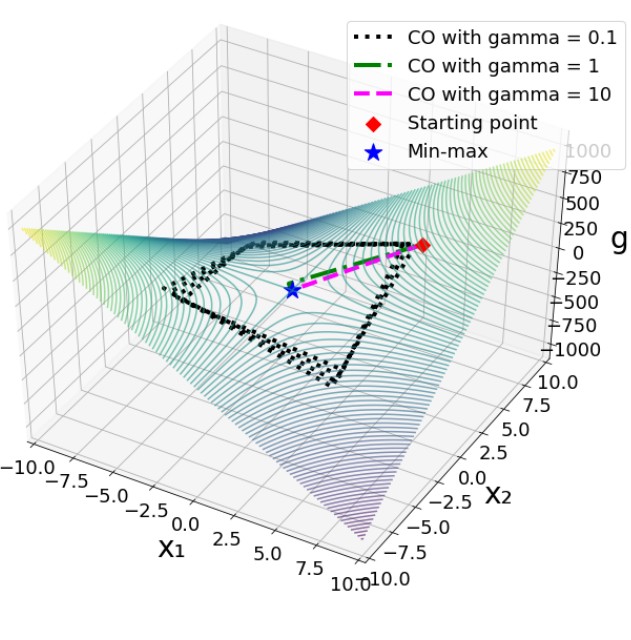

(b)

Figure 11: CO for 15 iterations with different values of $\gamma$ for $g(x_1, x_2) = f(x_1) + cx_1x_2 - f(x_2)$ where $f(x) = \log(1 + e^x)$ and $c = 10$. The $\gamma = 0.1$ curve makes a cyclic pattern around the min-max, while the other curves go directly to the min-max.

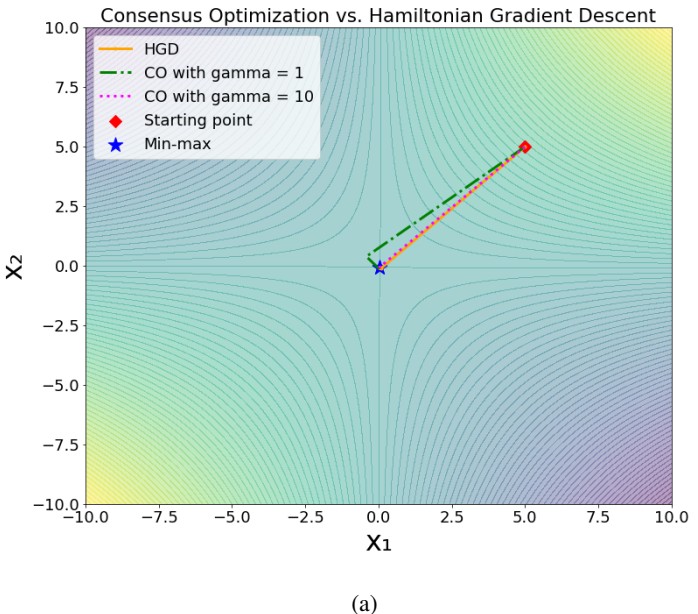

(a)

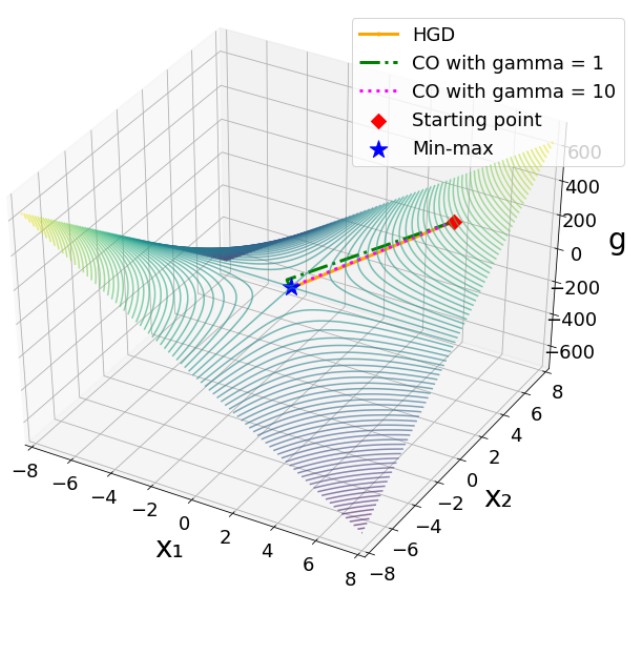

(b)

Figure 12: HGD vs. CO for 15 iterations with different values of $\gamma$ for $g(x_1, x_2) = f(x_1) + cx_1x_2 - f(x_2)$ where $f(x) = \log(1 + e^x)$ and $c = 10$.

## K.2 NONCONVEX-NONCONCAVE OBJECTIVE

The nonconvex-nonconcave objective we use is $g(x_1, x_2) = F(x_1) + cx_1x_2 - F(x_2)$ where $F$ is defined as in (16) in Appendix E.

$$F(x) = \begin{cases} -3(x + \frac{\pi}{2}) & \text{for } x \leq -\frac{\pi}{2} \\ -3\cos x & \text{for } -\frac{\pi}{2} < x \leq \frac{\pi}{2} \\ -\cos x + 2x - \pi & \text{for } x > \frac{\pi}{2} \end{cases} \tag{73}$$

We show a plot of $F$ in Figure 13.

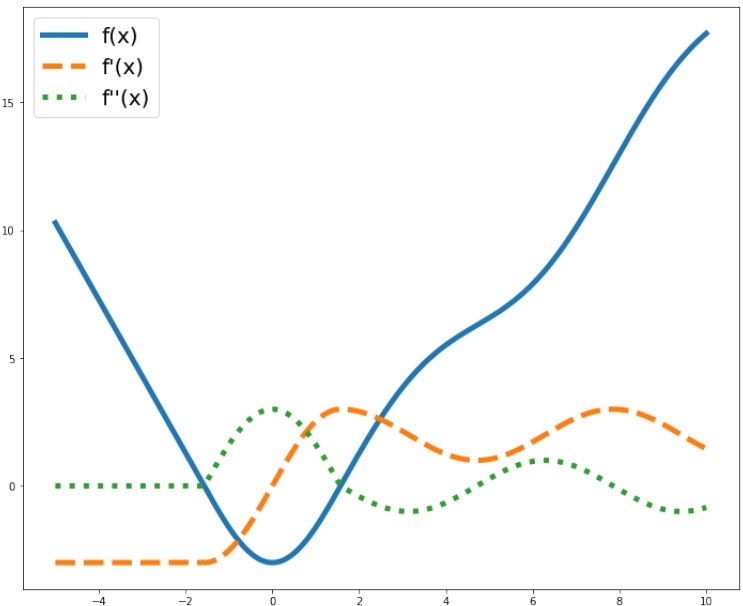

Figure 13: Plot of nonconvex function $F(x)$ defined in (16), as well as its first and second derivatives

As in the convex-concave case, when $c = 3$, SGDA converges, and when $c = 10$, SGDA diverges. Again, HGD and CO (for large enough $\gamma$) tend to converge faster when $c$ is larger.

### K.2.1 SGDA CONVERGES ($c = 3$)

These plots show $g$ when $c = 3$, so SGDA converges, as does CO with $\gamma = 0.1$.

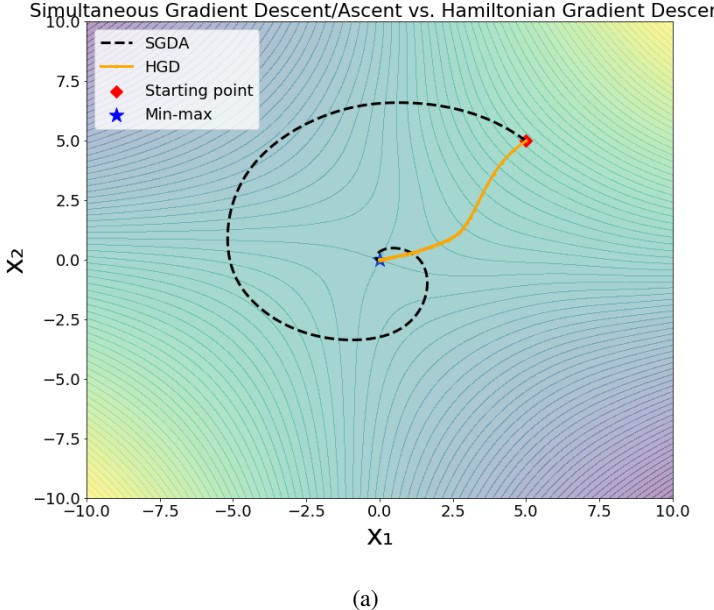

(a)

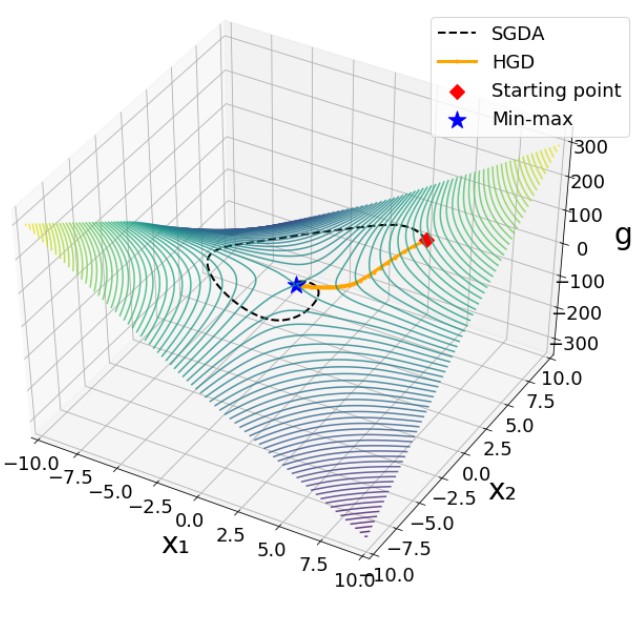

(b)

Figure 14: SGDA vs. HGD for 300 iterations for $g(x_1, x_2) = F(x_1) + cx_1x_2 - F(x_2)$ where $F(x)$ is defined in (73) and $c = 3$. SGDA slowly circles towards the min-max, and HGD goes more directly to the min-max.

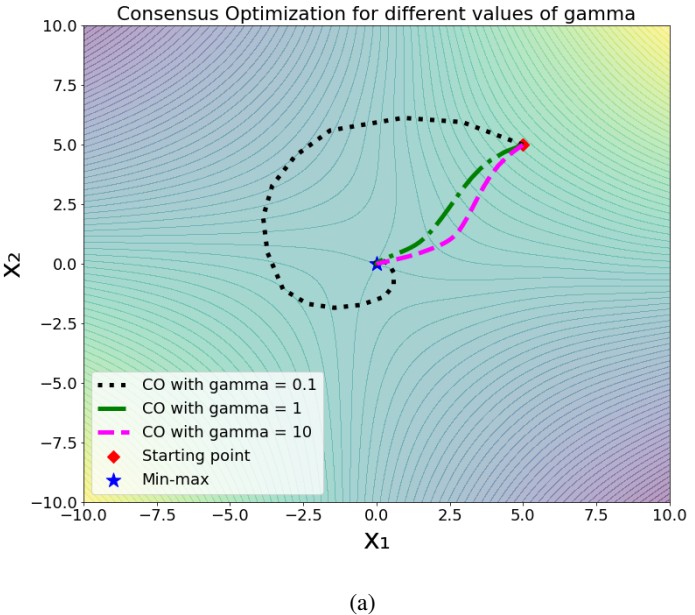

(a)

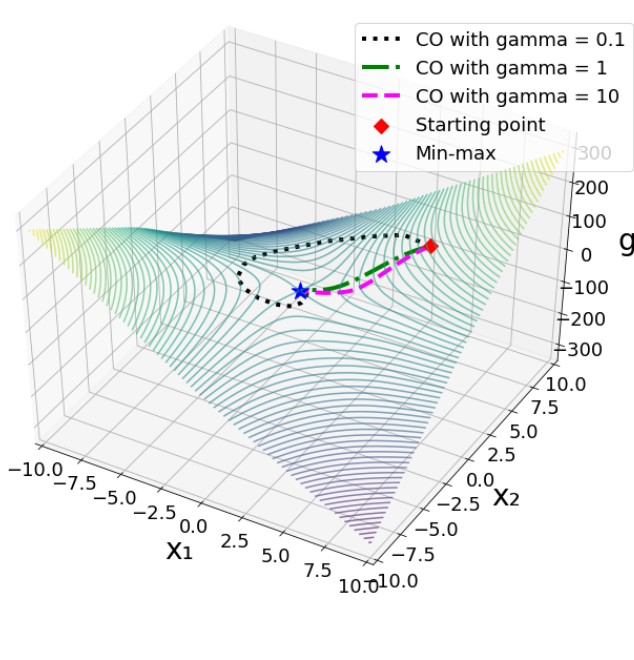

(b)

Figure 15: CO for 100 iterations with different values of $\gamma$ for $g(x_1, x_2) = F(x_1) + cx_1x_2 - F(x_2)$ where $F(x)$ is defined in (73) and $c = 3$. The $\gamma = 0.1$ curve slowly circles towards the min-max, while the other curves go more directly to the min-max.

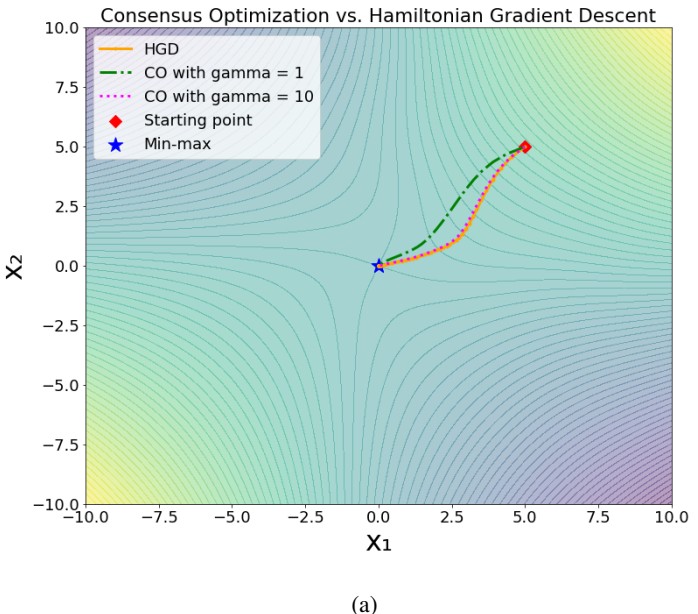

(a)

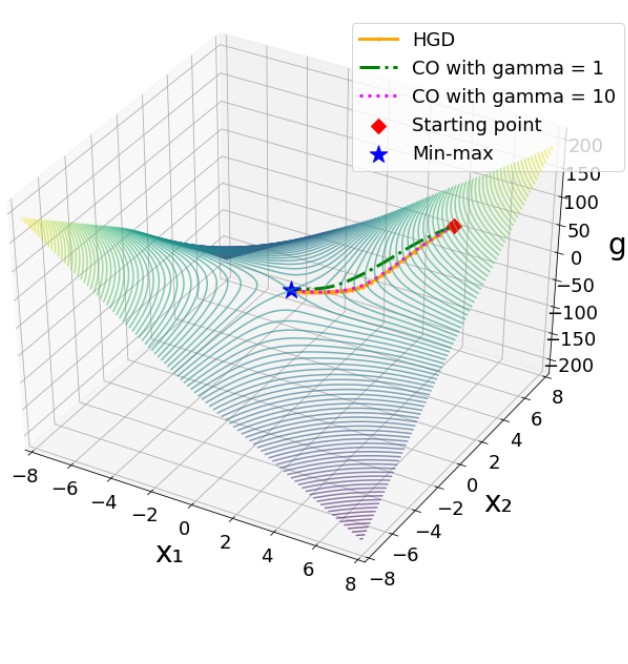

(b)

Figure 16: HGD vs. CO for 100 iterations for $g(x_1, x_2) = F(x_1) + cx_1x_2 - F(x_2)$ where $F(x)$ is defined in (73) and $c = 3$ with different values of $\gamma$.

### K.2.2 SGDA DIVERGES ($c = 10$)

These plots show $g$ when $c = 10$, so SGDA diverges, as does CO with $\gamma = 0.1$. Note that in this case, CO with $\gamma \geq 1$ and HGD both require very few iterations (typically about 2) to reach the min-max.

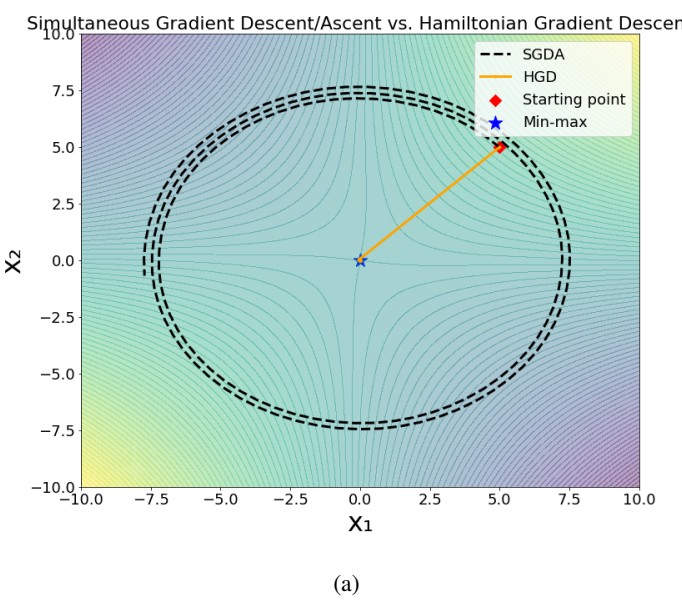

(a)

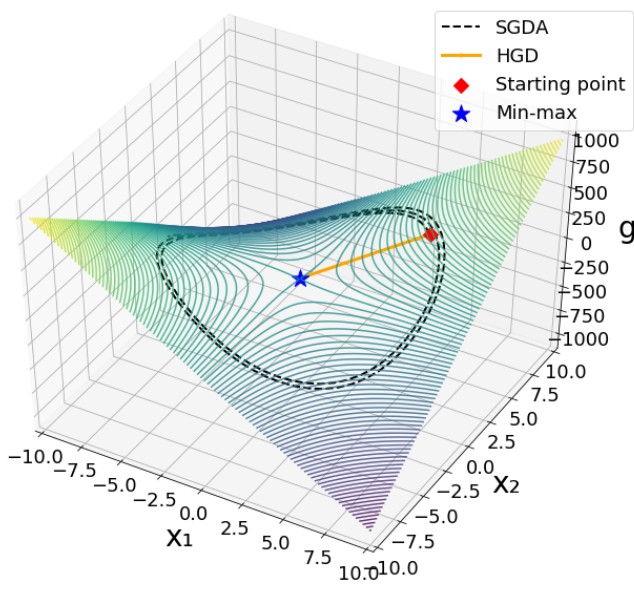

(b)

Figure 17: SGDA vs. HGD for 150 iterations for $g(x_1, x_2) = F(x_1) + cx_1x_2 - F(x_2)$ where $F(x)$ is defined in (73) and $c = 10$. SGDA slowly circles away from the min-max, while HGD goes directly to the min-max.

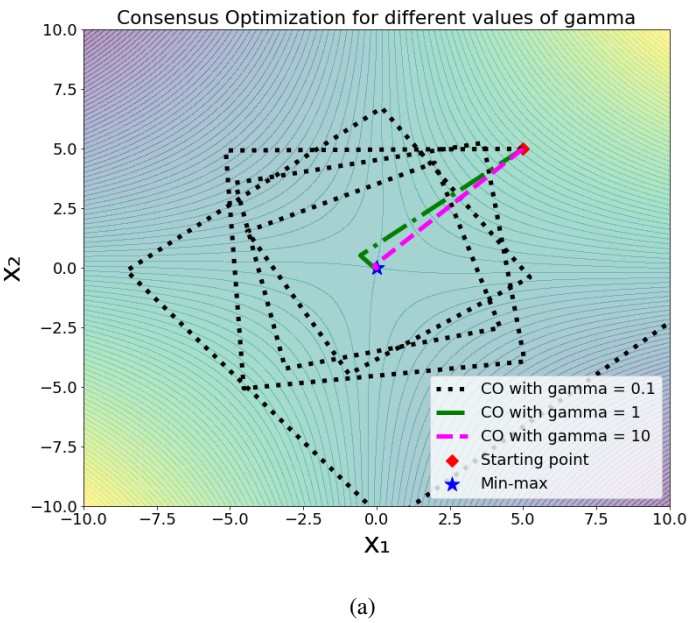

(a)

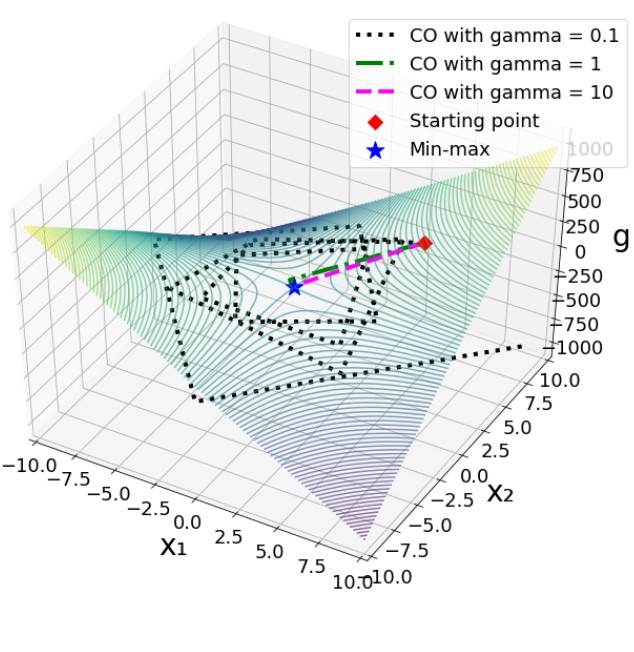

(b)

Figure 18: CO for 15 iterations with different values of $\gamma$ for $g(x_1, x_2) = F(x_1) + cx_1x_2 - F(x_2)$ where $F(x)$ is defined in (73) and $c = 10$. The $\gamma = 0.1$ curve makes an erratic cycle around the min-max, slowly diverging, while the other curves go directly to the min-max.

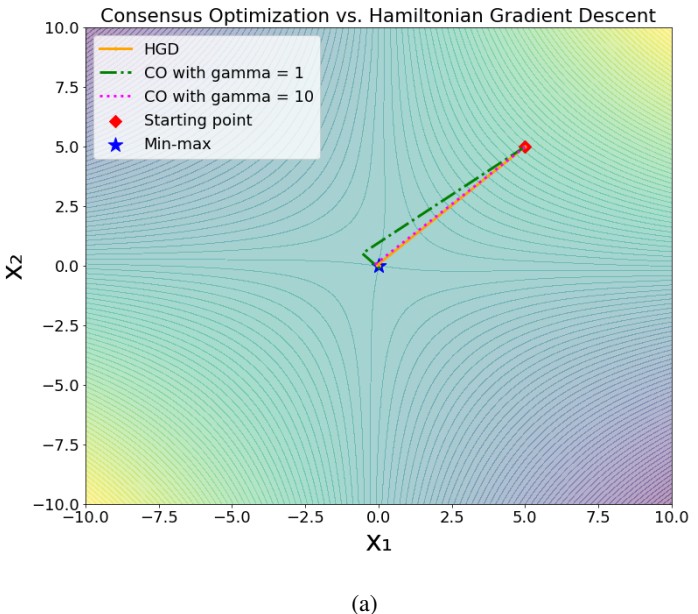

(a)

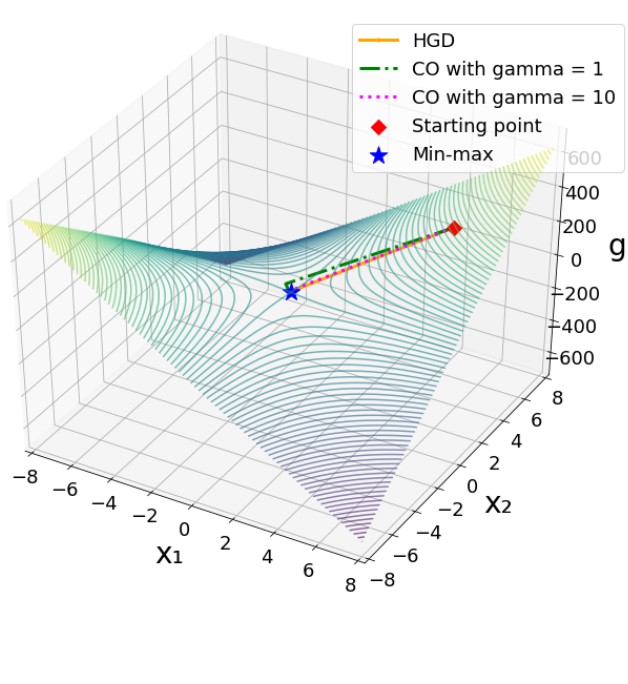

(b)

Figure 19: HGD vs. CO for 15 iterations with different values of $\gamma$ for $g(x_1, x_2) = F(x_1) + cx_1x_2 - F(x_2)$ where $F(x)$ is defined in (73) and $c = 10$.

### K.3 Convergence of HGD for nonconvex-nonconvex objective with different-sized bilinear terms

In this section, we look at the convergence of HGD for the same objective as discussed in the previous section, namely $g(x_1, x_2) = F(x_1) + cx_1x_2 - F(x_2)$ where $F$ is defined as in (16) in Appendix E.

$$F(x) = \begin{cases} -3(x + \frac{\pi}{2}) & \text{for } x \leq -\frac{\pi}{2} \\ -3\cos x & \text{for } -\frac{\pi}{2} < x \leq \frac{\pi}{2} \\ -\cos x + 2x - \pi & \text{for } x > \frac{\pi}{2} \end{cases} \tag{74}$$

In this case, we will vary $c$ to show that HGD converges faster for higher $c$ and will not converge for sufficiently low $c$.

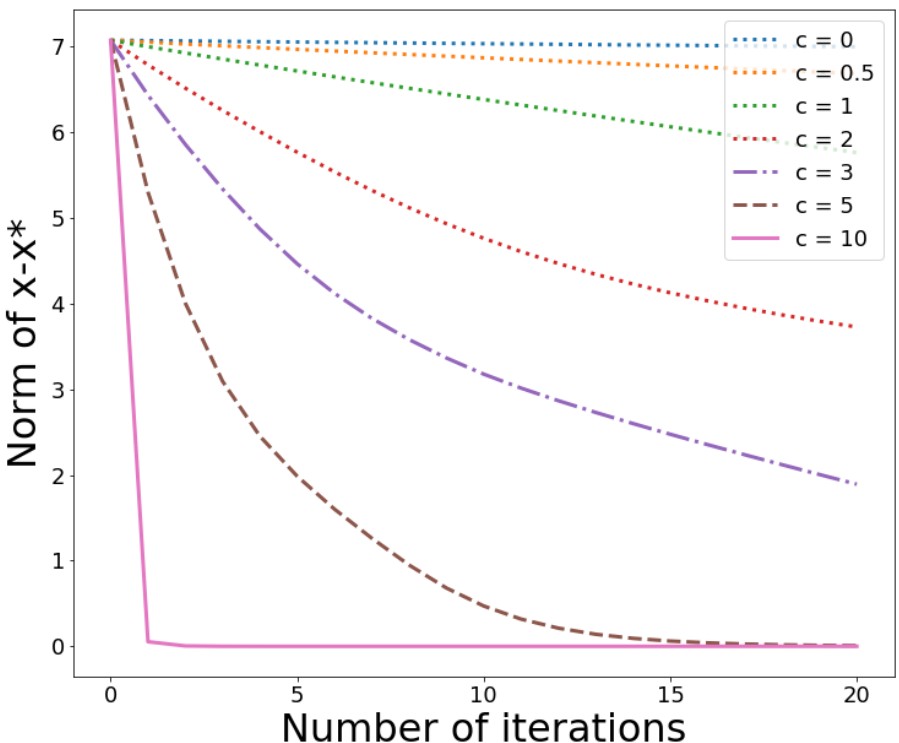

Figure 20: Distance to minmax for HGD iterates for different values of $c$ in the objective $g(x_1, x_2) = F(x_1) + cx_1x_2 - F(x_2)$ where $F(x)$ is defined in (73).

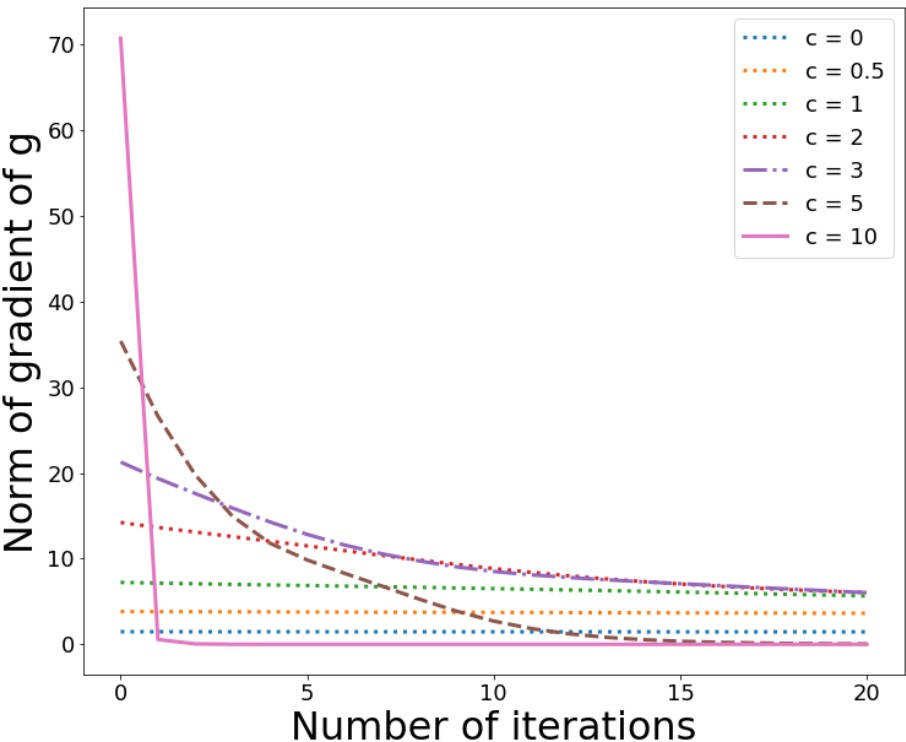

Figure 21: Gradient norm for HGD iterates for different values of $c$ in the objective $g(x_1, x_2) = F(x_1) + cx_1x_2 - F(x_2)$ where $F(x)$ is defined in (73). Since all runs are initialized at $(5, 5)$, when $c$ is increased, the initial gradient norm also increases. Nonetheless, HGD still converges faster for the cases with higher $c$.

