# OpenReview forum: "Last-iterate convergence rates for min-max optimization"
_ICLR.cc/2020/Conference — Reject_

### Official Review · AnonReviewer2 · 2019-10-22
**Official Blind Review #2**

**Rating:** 6

**Review:**

This paper shows that Hamiltonian gradient descent (HGD), which is gradient descent on the norm of the squared norm of the vector field, achieves linear convergence for a broader range of problems than bilinear and convex-strongly concave formulations. In particular, the authors show the result for convex-concave problems satisfying a “sufficiently bilinear” condition that is related to the PL conditions. Finally, the authors argue that consensus optimization (CO) can be viewed as a perturbation of HGD when the parameter choice is big enough. From this viewpoint they derive convergence rates for CO on the broader set of problems. This provides some further theoretical justification of the success of CO on large scale GAN problems.

The paper is presented in a clear manner, with the objectives and analysis techniques delineated in the main paper. This was helpful to get a sense of the main points before going through the appendix. The objective of the paper is to extend the problem settings for which there is last iterate min-max convergence rates, which now exist for bilinear, strongly convex-strongly concave, and convex-strongly concave problems. The authors achieve this by analyzing HGD and giving convergence rates for when a “sufficiently bilinear condition is satisfied”. The primary idea behind the proof techniques is to show that the objective (Hamiltonian) satisfies the PL condition. I found this to be an interesting approach.

As a result, the main question in evaluating this paper is on the significance of the result and the generality of the “sufficiently bilinear” condition. I tend to lean toward the result carrying some significance since it does extend the class of problems for which the convergence rates exists. However, the weakness is that the condition is opaque and it is not entirely clear how broad of class of problems this condition would apply to. I do acknowledge that the authors did a reasonable job of trying to clear this up in section 3.2 and section G of the appendix. It did still leave me wanting more with respect to the practical significance though.

Finally, I found the connection to CO valuable. In particular, since this paper does not show large-scale experiments, the connection serves to provide some more theoretical evidence for they CO performs well in practice.

Post Author Response: Thanks for the response. I agree with your perspective and think this paper should be accepted.

**Experience Assessment:**

I have read many papers in this area.

**Review Assessment: Checking Correctness Of Derivations And Theory:**

I assessed the sensibility of the derivations and theory.

**Review Assessment: Checking Correctness Of Experiments:**

I assessed the sensibility of the experiments.

**Review Assessment: Thoroughness In Paper Reading:**

I read the paper at least twice and used my best judgement in assessing the paper.

---

> ### Author Response · Authors · 2019-11-13
> **Thanks for the feedback!**
>
> We thank the reviewer for the comments and suggestions. Our paper indeed focuses on theoretical results, but we believe the theory has some practical implications as well. In particular, while the exact form of the sufficiently bilinear condition may be somewhat unwieldy, the result gives concrete evidence that having higher bilinearity can aid convergence for certain algorithms, even for settings that are not purely bilinear. This indicates that one should pay attention to the magnitude and condition number of the off-diagonal of the Jacobian when constructing a min-max problem and choosing an algorithm to solve the problem.

---

### Official Review · AnonReviewer3 · 2019-10-23
**Official Blind Review #3**

**Rating:** 6

**Review:**


Summary:
The paper, considers methods for solving smooth unconstrained min-max optimization problems.  In particular, the authors prove that the Hamiltonian Gradient Descent (HGD) algorithm converges with linear convergence rate to the min-max solution. One of the main contributions of this work is that the proposed analysis is focusing on last iterate convergence guarantees for the HGD. This result, as the authors claim can be particularly useful in the future for analyzing more general settings (nonconvex-nonconcave min-max problems).
In addition, two preliminary convergence theorems were provided for two extensions of HGD: (i) a stochastic variant of HGD and (ii)  Consensus Optimization Algorithm (CO) (by establishing connections of CO and HGD).

Main Comments:
The paper is well written and the main contributions are clear. I believe that the idea of the paper is interesting and the convergence analysis seems correct, however i have some concerns regarding  the presentation and the combination of different assumptions used in the theory.

1) I think definition 2.5 of Higher order Lipschitz is very strong assumption to have. What exactly means? Essentially the authors upper bounded any difficult term appear in the theorems. Is it possible to avoid having something so strong? Please elaborate.

2) In assumption 3.1 is not clear what $L_H$ is. This quantity never mentioned before. Reading the Lemmas of Section 4 (Lemma 4.4) you can see that it is the smoothness parameter of function $H$. Thus, is not necessary to have it there (not important for the definition).

3) What is the main difference on the combination of assumptions on Theorems 3.2, 3.2 and 3.4. Which one is stronger. Is there a reason for the existence of Theorem 3.3?

4) All the results heavily depend on the PL condition. I think having this in mind, showing the convergence of Theorems 3.2-3.4 is somehow trivial. In particular, one can propose several combinations of assumptions in order for the function H to satisfy the PL condition. Can we avoid having the PL condition? The authors need to elaborate more on this.

5) In Theorem 5.2, the term 1/sqrt(2) is missing from the final bound.

Minor Suggestions:
In first paragraph of page 5 where the authors divide the existing literature into the three particular cases, I am suggesting to add the refereed papers inside each one of this cases (which papers assumed function g bilinear , which papers strongly convex-concave etc.)

I understand that the main contribution of the work is the theoretical analysis of the proposed method but would like to see some numerical evaluation in the main paper. There are some preliminary results in the appendix but it will be useful for the reader if there are are some plots showing the benefit of the method in comparison with existing methods that guarantee convergence (which method is faster?). In the current experiments there is a comparison only with CO algorithm and SGDA.

In general i find the paper interesting, with nice ideas and I believe that will be appreciated from researchers that are interested on smooth games and their connections to machine learning applications.

I suggest weak accept but I am open to reconsider in case that my above concerns are answered.

**********after rebuttal********
I would like to thank the authors for their reply and for the further clarification.
I will keep my score the same but I highly encourage the authors to add some clarification related to my last comment on the globally bounded gradient.
In their response they mentioned that the analysis only requires that  H is smooth and that $\|\xi(x^{(0)})\|$ is sufficient bound. This needs to be clear in the paper (add clear arguments and related references).
In addition, in their response they highlight the non-increasing nature of function H over the course of the algorithm which is important for their argument. Having this in mind note that the theoretical results on stochastic variant presented in the paper are wrong. In SGD,  function H does not necessarily decrease over the course of the algorithm. The authors either need to remove these results or restate them in a different way in order to satisfy the assumed conditions.

**Experience Assessment:**

I have read many papers in this area.

**Review Assessment: Checking Correctness Of Derivations And Theory:**

I carefully checked the derivations and theory.

**Review Assessment: Checking Correctness Of Experiments:**

I assessed the sensibility of the experiments.

**Review Assessment: Thoroughness In Paper Reading:**

I read the paper thoroughly.

---

> ### Author Response · Authors · 2019-11-13
> **Thanks for the feedback!**
>
> We thank the reviewer for the comments and suggestions. We address points individually below:
>
> 1) The Higher-order Lipschitz condition is necessary for us to use the PL convergence guarantee. This condition is similar to assumptions made in convex optimization, especially where higher-order updates are involved (see eg. Agarwal et al. 2017 and Bubeck et al. 2019). If the iterates of the algorithm always have bounded norm (eg. due to constraints or regularization), then three-times differentiable functions will satisfy the Higher-order Lipschitz condition for our purposes. This is because it suffices for the condition to hold for only the iterates of the algorithm ($x^{(1)},x^{(2)},...$), rather than for all of $\mathbb{R}^d$.
>
> 2) We thank the reviewer for this remark. We wanted $L_H$ to be defined for our theorem statements, but we can see how it is confusing as is. We will make it clear that $L_H$ is the smoothness parameter of $H$.
>
> 3) Theorem 3.4 holds in the broadest setting out of all of these results. Theorems 3.2 and 3.3 have slightly tighter bounds for their respective settings. We will clarify this in the surrounding text.
>
> 4) It is true that these results rely on the PL condition, and this is unavoidable for our current results. The novel perspective in this paper is that we consider the PL condition on a different objective, namely the squared gradient norm, rather than on the game objective $g$. This perspective allows us to prove our new bounds, although we still require some nontrivial linear algebra. The PL condition also allows us to easily prove our stochastic HGD results.
>
> 5) The 1/sqrt(2) should cancel out on both sides of the guarantee in Theorem 5.2 (and eg. in equation 68).
>
> Minor suggestions:
> -We appreciate the suggestion for page 5 and will make this change in the final version.
>
> We thank the reviewer for recognizing our theoretical contributions. We would be happy to include some further experiments in the final version comparing HGD with other algorithms such as extragradient.
>
>
> References:
> Agarwal, Naman, and Elad Hazan. "Lower bounds for higher-order convex optimization." COLT 2018.
>
> Bubeck, Sébastien, et al. "Near-optimal method for highly smooth convex optimization." COLT 2019.

---

> > ### Comment · AnonReviewer3 · 2019-11-15
> > **Clarification on main assumptions**
> >
> > Thanks for the response.
> >
> > I still have some concerns regarding the assumptions used in the proofs.
> >
> > I agree with the authors that the Higher-order Lipschitz condition  is similar to assumptions made previously in convex optimization. However, it was shown recently that for unconstrained optimization problems having strong convexity and bounded gradient is unrealistic, leading to an empty set of functions (see discussions in [1] and [2]).
> > Note also that the PL condition can be seen as a relaxation of strong convexity. Having this in mind can the authors provide an example of a function that satisfy the PL condition and also have bounded gradients?
> >
> > Note that using the Higher-order Lipschitz condition the hamiltonian function H(x) (the PL function) has bounded gradient:
> > $\|\nabla H(x) \| \leq \| \xi \| \| J\| \leq L_1 L_2$
> >
> > In addition in the proof of Lemma 4.4 can the authors elaborate more on what inequality/assumption they use to upper bound $\| \xi(u)-\xi(v) \|$ (last step of the proof)?
> >
> > Mentioned references:
> > [1] Nguyen, Lam, et. al "SGD and Hogwild! Convergence Without the Bounded Gradients Assumption." ICML, pp. 3747-3755. 2018.
> > [2] Gower, R.M., Loizou, N., Qian, X., Sailanbayev, A., Shulgin, E. and Richtárik, P., "SGD: General Analysis and Improved Rates". ICML, pp. 5200-5209, 2019

---

> > > ### Author Response · Authors · 2019-11-15
> > > **Thanks for the additional feedback!**
> > >
> > > The reviewer raises a valid point, and this warrants some additional clarification in the paper. While there will not be functions that satisfy the PL condition and have globally bounded gradient, our analysis only requires that $H$ is smooth over the iterates of the algorithm. Since $H = \frac{1}{2}\| \xi\|^2$ and $H$ is non-increasing over the course of the algorithm (due to GD being a descent method), we will have $\|\xi(x^{(k)})\| \le \|\xi(x^{(0)})\|$ for $k \ge 0$. Thus, it suffices to use $\|\xi(x^{(0)})\|$ as our bound on $\| \xi\|$ in our theorems. As such, we don’t need to assume $\|\xi\|$ is globally bounded.
> > >
> > > We touched on the above very briefly in Appendix E, but we will clarify this in the main body in the final version. We have example functions in Appendix E and Appendix K.1 that satisfy our remaining assumptions.
> > >
> > >
> > > For the proof of Lemma 4.4, we use the fact that $\|\nabla \xi\|$ is bounded, which implies that $\xi$ is Lipschitz. This can be seen from the fundamental theorem of calculus for line integrals, which gives:
> > > $$\xi(u) - \xi(v) =  \int_0^1 \nabla \xi((1-t)v + tu) dt \cdot (u-v)$$
> > > Since $\|\nabla \xi \|$ is bounded, we get the result.

---

### Official Review · AnonReviewer1 · 2019-10-23
**Official Blind Review #1**

**Rating:** 6

**Review:**

*Summary*

This paper study the convergence of Hamiltonian gradient descent (HGD) on minmax games. The paper show that under some assumption on the cost function of the min max that are (in some sense) weaker than strong convex-concavity. More precisely, they use the ‘bilinearity’ of the objective (due to the interaction between the players) to prove that the squared norm of the vector field of the game follows some Polyak Lojasiewicz condition. Thus the proof is concluded by the linear (resp. sublinear) convergence of gradient descent (resp. stochastic GD) under PL assumption.

*Decision*

I think that is work is clearly very interesting. The fact to prove linear convergence rate without strong-convex-concavity is quite surprising. And this paper brings nice tools to analyse HGD. Also the result on Stochastic HGD is very interesting.

However, I am wondering whether this paper is perfectly suited to ICLR conference due to the lack of experiment, practical implication given by the theory, or theory in the non-convex setting (I know that the latter is a huge open question and I am not criticizing the absence of theory in the non-convex-concave setting).
One way to improve to work would be to provide practical takeaways from the theory or to provide experiments in the main paper.

Regarding the practical limitation of this work:
- the sufficient bilinearity condition are hard to meet in practice. (even for convex-concave problems)
- In a non-convex-concave setting, Hamiltonian gradient descent is attracted the any stationary point, even “local maxima” (or the equivalent in the minmax setting). Making this algorithm not very practical. (However, CO is)

However, I really think that the community is currently lacking of understanding on minmax optimization and that we need better training method in many practical emergent frameworks that are minmax (such as GANs or multi agent learning). That is why, I would vote for a weak accept.

*Questions*
- What are the practical implication of your work ? for instance does it say anything on how to tune $\gamma$ for CO ?

*Remarks*
- It is claimed that Theorem 3.4 gives the first linear convergence rate for minmax that does not require strong-convex or linearity. Note that, recently [1] seem to propose a result on extragradient in the same vein (i.e. without strong convexity or linearity).
- (Minor) $\alpha$ not alway have the same unit: Thm 3.2 it is proportional to a strong convexity and in Lemma 4.7 it is proportional to a strong convexity squared (actually the PL of the squared norm of the gradient). For clarity it might be interesting to use the notation $\alpha^2$ in Lemma 4.7. The same way for unit consistency I would use $L_H^2$ instead of $L_H$

[1] Azizian, Waïss, et al. "A Tight and Unified Analysis of Extragradient for a Whole Spectrum of Differentiable Games." arXiv preprint arXiv:1906.05945 (2019).


=== After rebuttal ===
I've read the authors's response.
The concern raised by reviewer 3 is very important. The descent lemma used by the author is not valid for the stochastic result. The authors should address that in their revision.
I however maintain my grade.



**Experience Assessment:**

I have published in this field for several years.

**Review Assessment: Checking Correctness Of Derivations And Theory:**

I carefully checked the derivations and theory.

**Review Assessment: Checking Correctness Of Experiments:**

I assessed the sensibility of the experiments.

**Review Assessment: Thoroughness In Paper Reading:**

I read the paper thoroughly.

---

> ### Author Response · Authors · 2019-11-13
> **Thanks for the feedback!**
>
> We thank the reviewer for the comments and suggestions. As the reviewer points out, the community currently lacks a strong theoretical understanding of minmax optimization, and we believe our work helps to fill this gap. We comment on the practical implications of our work below:
>
> 1) While the exact form of the sufficiently bilinear condition may be somewhat unwieldy, the result gives concrete evidence that having higher bilinearity can aid convergence for certain algorithms, even for settings that are not purely bilinear. This indicates that one should pay attention to the magnitude and condition number of the off-diagonal of the Jacobian when constructing a min-max problem and choosing an algorithm to solve the problem.
>
> 2) In non-convex-concave settings, HGD will converge to all types of stationary points, as the reviewer points out. We propose some modifications to HGD to allow it to work in non-convex settings in Appendix A, which essentially amount to explicitly determining the local curvature of the problem and running a modified algorithm, such as Hamiltonian Gradient Ascent, near undesirable critical points. This would allow us to show similar local convergence guarantees to those proven by other works in the area (see Appendix A). However, as the reviewer points out as well, the HGD analysis is also useful because it implies similar convergence results for CO, which is a practical algorithm.
>
> 3) Our result for CO shows that as long as $\gamma \ge 4L_g/\alpha$, then CO will converge in sufficiently bilinear settings (currently it’s written as $\gamma = 4L_g/\alpha$ but we will change this in the final version). This indicates that increasing $\gamma$ may speed up convergence when we are in a sufficiently bilinear region (and in particular, the algorithm may not converge if $\gamma$ is too small and the region has a very large bilinear term). If $\gamma$ is too large, CO will converge to stationary points that are not local min-maxes, so these two phenomena must be traded off. One could potentially detect which regime one is in by computing a few eigenvalues of the Jacobian (using a logarithmic number of Hessian-vector products) during or after training.
>
> Other comments:
> -We thank the reviewer for pointing out Azizian et al. 2019. This work was released concurrently to ours on Arxiv and indeed seems to have some similar findings. We will include a reference to it in our revised version.
> -We thank the reviewer for the comment on notation and will incorporate it into the final version.
>
> References:
> Azizian, Waïss, et al. "A Tight and Unified Analysis of Extragradient for a Whole Spectrum of Differentiable Games." arXiv preprint arXiv:1906.05945 (2019).

---

### Decision · Program_Chairs · 2019-12-19

**Decision:**

Reject

**Comment:**

This provides a simple analysis of an existing algorithm for min-max optimization under some favorable assumptions.  The paper is clean and nice, though unfortunately lands just below borderline.

I urge the authors to continue their interesting work, and amongst other things address the reviewer comments, for example those on stochastic gradient descent.